



**Impacts of anomalies in Arctic sea ice outflow on sea ice in the Barents and Greenland**
**Seas during the winter-to-summer seasons of 2020**
Fanyi Zhang[1,2], Ruibo Lei[2,1][*], Xiaoping Pang[1], Mengxi Zhai[2], Na Li[2]
[1]Chinese Antarctic Center of Surveying and Mapping, Wuhan University, Wuhan 430079, China
[2]Key Laboratory for Polar Science of the MNR, Polar Research Institute of China, Shanghai 200136, China
*Correspondence to:* Ruibo Lei(leiruibo@pric.org.cn.)
**Abstract:** Arctic sea ice outflow to the Atlantic Ocean is essential to Arctic sea ice mass loss and the hydrographical and
ecological environments in the Barents and Greenland Seas (BGS). In the context of the extremely positive Arctic Oscillation
(AO) in January–March 2020, the impacts and feedback mechanisms on a seasonal scale of anomalies in Arctic sea ice outflow
on winter–spring sea ice and other marine environmental conditions in the subsequent months until early summer in the BGS
were investigated. The results reveal that the total sea ice area flux (SIAF) through the Fram Strait, the Svalbard-Franz Josef
Land, and the Franz Josef Land-Novaya Zemlya passageways in January–March and June 2020 were higher than the 1988–
2020 climatology, mainly through the Fram Strait (77.6%). The interannual variability of this total SIAF was dominated by
changes in ice motion speed ($R$ = +0.86, $P$ < 0.001). The relatively high ice speed along the Transpolar Drift in January–June
2020 was related to the positive phases of winter (JFM) AO and the winter-spring air pressure gradient across the western and
eastern Arctic Ocean. The abnormally high Arctic sea ice outflow led to an increased sea ice area and thickness in the BGS,
which has been observed since March 2020, especially in May–June. In this region, the April sea ice area was significantly
negatively correlated with synchronous sea surface temperature (SST) as well as the lagging SST of 1–3 months. High sea ice
area in spring (AMJ) 2020 also inhibited phytoplankton bloom, with an extremely low *Chlorophyll-a* concentration observed
over the BGS in April. Therefore, this study suggests that winter–spring Arctic sea ice outflow can be considered as a predictor
of changes in sea ice and other marine environmental conditions in the BGS in the subsequent months, at least until early
summer. The results increase our understanding of the physical connection between the central Arctic Ocean and the peripheral
seas.
**KEYWORDS**: Arctic Ocean; Sea ice; Transpolar Drift; Atmospheric circulation pattern; Barents Sea; Greenland Sea



## 1. Introduction

Arctic sea ice has been experiencing a dramatic loss over the past four decades, and the overall decline in sea ice extent is statistically significant in all seasons (Cavalieri and Parkinson, 2012). In winter, due to the absence of land constraints, reductions in the Arctic sea ice extent occurred mainly in the peripheral seas, particularly in the Barents and Greenland Seas (BGS). From 1979 to 2016, sea ice changes in the Barents and Greenland Seas accounted for 27% and 23% of the total Arctic sea ice extent loss in March, respectively (Onarheim et al., 2018). Changes in Arctic sea ice may have potentially far-reaching effects not only on Arctic local climate and ecological environments but also on extreme weather or climatic events at lower latitudes (Schlichtholz, 2019). Previous studies have revealed the relations of Eurasian winter cold anomalies to sea ice reduction in the Barents Sea (e.g., Mori et al., 2014).

Through the regulations of thermodynamic and dynamic processes, large-scale atmospheric circulation patterns have significant implications on sea ice growth and decay, as well as advection and spatial redistribution (Frey et al., 2015; Dorr et al., 2021; Dethloff et al., 2022). Dynamically, enhanced wind forcing, associated with anomalous atmospheric circulations, could induce increased sea ice motility and deformation, especially for Arctic sea ice outflow through the Fram Strait (e.g., Cai et al., 2020). Associated with the conveyor belt of the Transpolar Drift (TPD), Arctic sea ice can be exported to the BGS and finally enter the North Atlantic (Kwok, 2009), which is an important mechanism for decreases in the total sea ice volume over the pan-Arctic (Smedsrud et al., 2017), especially for the loss of multi-year ice (Kwok et al., 2009). Moreover, Arctic sea ice advection along the TPD is also capable of transporting ice-rafted materials or extend ice-associated biomes from the Eurasian shelf to the Arctic basin, and eventually out of the Arctic Ocean (Mørk et al., 2011; Peeken et al., 2018; Krumpen et al.,2020). The Arctic sea ice outflow, associated with equivalent fresh water outflow being comparable to that carried by the East Greenland current (Spreen et al., 2009; de Steur et al., 2014), contributed significantly to the formation of deep water in the north of the Atlantic Ocean (Lemke et al., 2000). In turn, the increase in the oceanic heat inflow from the north Atlantic Ocean leads to the Atlantification and promotes the retreat of sea ice in the BGS (Shu et al., 2021).

As the peripheral seas of the Arctic Ocean, the BGS are not completely covered by sea ice even in winter, so the ocean dynamic processes and atmosphere-ocean interactions are relatively strong in this region compared to the central Arctic Ocean (Smedsrud et al., 2013). Sea ice outflow from the Arctic Ocean plays a crucial role in shaping the icescape in this region. And most notably, more marine primary productivities occur in the BGS than in other regions for the waters north of the Arctic Circle due to the supply of nutrients from the south and the availability of more photosynthetic light because of the relatively low sea ice coverage (Arrigo and van Dijken, 2015; Mayot et al., 2020). Naturally, the bloom of primary productivity in this region is greatly affected by the distribution and seasonality of sea ice, mainly by regulating the available photosynthetic light



(Wassmann et al., 2010). Thus, further revealing the influence and feedback mechanisms of abnormal Arctic sea ice outflow
on the marine environmental conditions in the downstream of TPD over the BGS on a seasonal scale could improve the
understanding of the physical connections between the central Arctic Ocean and the BGS, which is still not particularly clear,
especially in conjunction with some extreme atmospheric circulation events.

Variations in Arctic sea ice outflow to the BGS are associated with a variety of large-scale atmospheric circulation patterns

and local synoptic events (Bi et al., 2016), among which the atmospheric circulation patterns of the Arctic Oscillation (AO)
(Kwok, 2009) and the Central Arctic west-east air pressure gradient Index (CAI; Vihma et al., 2012) can play significant roles.
The AO index is the dominant pattern of surface mean air pressure anomalies, with a positive AO index indicating below
normal air pressure in the Arctic and above normal over external regions (Dethloff et al., 2022). When the AO is in a relatively
extreme positive phase, the westward shift of the TPD allows thicker multi-year ice to be advected from the central Arctic
Ocean towards Fram Strait (Rigor et al., 2002). In January–March 2020, the AO experienced an unprecedented positive phase,
which led to the relatively rapid southward drift of the ice camp of the Multidisciplinary drifting Observatory for the Study of
Arctic Climate (MOSAiC) during the winter and early spring of 2019–2020 (Krumpen et al., 2021). The CAI, on the other
hand, represents the east-west gradient of the SLP across the central Arctic Ocean, approximately perpendicular to the TPD
(Vihma et al., 2012). Thereby, it can indicate the intensity of TPD to a high degree (Lei et al., 2016), which directly affects the
outflow of Arctic sea ice toward the BGS.

Thereby, the main objectives of this study are to clarify the effects of atmospheric circulation anomalies on Arctic sea ice

outflow during winter–spring 2020, and their effects on sea ice distributions and other marine conditions over the BGS in the
subsequent months until early summer, in order to reveal seasonal impacts and feedback mechanisms. The sections of this
paper are organized as follows. The datasets used to measure anomalies in atmospheric, sea ice, and oceanic conditions are
briefly described in Section 2. Section 3 presents the anomalies in atmospheric circulation and Arctic sea ice outflow in the
study year, as well as their influences on sea ice conditions in the BGS. Links of Arctic sea ice outflow to atmospheric
circulation, the impact of sea ice anomalies on the hydrographical and ecological conditions in the BGS, and the
representativeness of the connections between sea ice anomalies and the marine environments identified in 2020 related to the
climatological data, are discussed in Section 4. The conclusions are presented in the last section.
**2. Data and methods**
**2.1 Study area**

Our studies focus on the downstream region of the TPD, i.e., the Barents Sea (72°–80°N, 20°–60°E) and the Greenland



Sea (72°–80°N, 20°W–20°E) to assess the impacts of sea ice outflow from the Arctic Ocean on the sea ice and other marine
conditions in this region on a seasonal scale. To quantify the sea ice outflow, we calculated the sea ice area flux (SIAF) through
the passageways between the Arctic Ocean and the BGS region, i.e., through the Fram Strait, the Svalbard-Franz Josef Land
(S-FJL) passageway, and the Franz Josef Land-Novaya Zemlya (FJL-NZ) passageways (Figure 1), with the widths of about
448, 284, and 326 km, respectively.

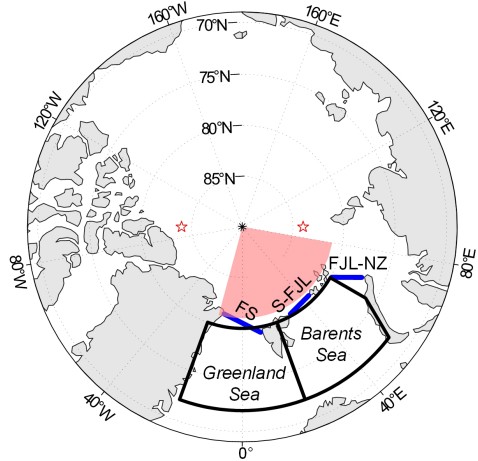


**Figure 1.** Geographical locations of the Barents and Greenland Seas (black frames). Blue lines represent the passageways defined for the
calculations of sea ice area flux. The red stars indicate the locations (90° W, 84° N, and 90° E, 84°N) defined to calculate the Central Arctic
west-east air pressure gradient Index (CAI). The Atlantic sector of TPD is shaded in red and the sea ice motion speed in this region is used
to quantify the link to wind speed.
**2.2 Data**
We used the National Snow and Ice Data Center (NSIDC) Polar Pathfinder version 4 sea ice motion (SIM) vectors and
National Oceanic and Atmospheric Administration (NOAA)/NSIDC Climate Data Record passive microwave sea ice
concentration version 4 (SIC) (Tschudi et al., 2019; Meier et al., 2021) to calculate the SIAF from the Arctic Ocean to the BGS.
The choice of this SIM product was motivated by its spatial completeness and temporal continuance. The SIM product is the
most optimal interpolation merged result using satellite remote sensing data, buoy observations, and reanalyzed wind data
(Tschudi et al., 2020). This product provides daily ice drift components georeferenced to the Equal-Area Scalable Earth Grid
(EASE-Grid) with a spatial resolution of 25 km. The SIC product was a rule-based combination of SIC estimates from the
National Aeronautics and Space Administration (NASA) Team (NT) algorithm (Cavalieri et al., 1984) and NASA Bootstrap
(BT) algorithm (Comiso, 1986), derived from the Scanning Multichannel Microwave Radiometer (SMMR), Special Sensor
Microwave Imager (SSM/I), and Special Sensor Microwave Imager/Sounder (SSMIS) radiometers. Daily SIC fields were



gridded on a 25-km resolution polar stereographic grid. Both datasets are available from October 1978 to the present. However,
there is a gap in the SIC dataset from 3 December 1987 through 12 January 1988. The sea ice area (SIA) was defined as the
cumulative area of the waters covered by sea ice with the SIC above 15%. For the study region, we used the SIC data since
1979 to estimate the SIA abnormal from January to June in the study year of 2020. In addition, we used data from the NSIDC
Sea Ice Index version 3 (Fetterer et al., 2017) to obtain monthly SIA changes in the Northern Hemisphere in 2020.

The sea ice thickness (SIT) data used to characterize the sea ice conditions in the study region was mainly derived from

satellite remote-sensed observations, and supplemented by the modeling product in early summer. The remote-sensed SIT data
was created from the merged CryoSat-2 and Soil Moisture and Ocean Salinity (SMOS) observations, hereinafter referred to as
CryoSat-2/SMOS (Ricker et al., 2017). The CryoSat-2/SMOS dataset makes full use of the detectability of SMOS for thin sea
ice (<1.0 m) and the measurement capability of CryoSat-2 for thicker sea ice, which ensures obtaining a more comprehensive
product of SIT. Weekly CryoSat-2/SMOS SIT data were available on a 25-km EASE-Grid during the freezing season of
October to mid-April from 2010 to the present. During the ice melt season from May–June, we used the monthly SIT modeling
product obtained from the Pan-Arctic Ice Ocean Modeling and Assimilation System (PIOMAS; Zhang and Rothrock, 2003).
The PIOMAS is a coupled ice-ocean model assimilation system that has been extensively validated and compared with satellite,
submarine, airborne, and in situ observations, which has proved it can reproduce the observed sea ice thickness well (Zhang
and Rothrock, 2003; Schweiger et al., 2011; Stroeve et al., 2014; Wang et al., 2016). The monthly PIOMAS SIT is gridded on
a generalized orthogonal curvilinear coordinate system with an average resolution of 22 km. We regridded the monthly SIT
data on the 25-km EASE-Grid to maintain consistency with the CryoSat-2/SMOS SIT data. Here, we used the CryoSat-
2/SMOS SIT from December to April, and the PIOMAS SIT from May to June in 2011–2020 to estimate the anomaly in SIT
during the study year of 2020.

Sea surface temperature (SST) and *Chlorophyll-a* (*Chl-a*) could be used as the best proxies to indicate the physical state

and primary productivity over a basin scale (Siswanto, 2020), and can be easily obtained from satellite remote sensing. Thus,
we used these two variables for the period 2005–2020 to characterize the anomalies in the hydrographical and ecological
conditions over the BGS during the study year, respectively. The SST data was obtained from NOAA Daily Optimum
Interpolation SST High Resolution dataset version 2, which assimilated buoy and ship-based data, satellite SST data, and proxy
SST data in the ice-covered regions of the Arctic (Reynolds et al., 2007). This dataset is available on a regular grid of
0.25°×0.25°. The merged *Chl-a* ocean colour product is available from the Ocean Colour-Climate Change Initiative (OC-CCI)
project, which is derived from multiple ocean colour sensors (Sathyendranath et al., 2021). The *Chl-a* dataset has a monthly
temporal resolution and a spatial resolution of 4 km.
The fifth generation reanalysis ERA5 datasets from European Centre for Medium-range Weather Forecasts (ECMWF)
provide sea level pressure (SLP), 2-m air temperature, 10-m surface wind, as well as surface net heat fluxes of longwave
radiation, shortwave radiation, sensible heat, and latent heat (Hersbach et al., 2020). These variables, with about 30-km
horizontal and 1-h temporal resolutions, were used to identify anomalies in surface atmospheric conditions or forcing over the
study region. The ERA5 reanalysis uses a significantly more advanced 4D-var assimilation scheme, with improved
performance over the Arctic compared to ERA-Interim (Graham et al., 2019). We used the monthly AO index provided by
NOAA Climate Prediction Center (CPC), which was constructed by projecting a daily 1000 hPa height anomaly at the 20°N
poles onto the AO loading pattern (Thompson and Wallace, 1998). In addition, the hourly SLP data from the ERA5 reanalysis
were used to calculate the monthly CAI, defined as the difference between SLPs at 90° W, 84° N, and 90° E, 84°N.
**2.3 Methods**
The SIAF was defined as the magnitude of the SIA conveyed through a defined gate during a given period. According to
Kwok (2009), we estimated the monthly SIAF by accumulating the daily integral of the products between the gate-
perpendicular component of the SIM and SIC along the defined passageways. Positive (negative) values correspond to the
southward (northward) SIAF. Prior to the estimation of SIAF, we interpolated the SIC into the SIM projection and retrieved
the gate-perpendicular SIM components. According to the trapezoidal rule, the SIAF was estimated as follows:
$SIAF = \sum_{i=1}^{n} u_i C_i \Delta x$                                                                                 (1)
where $n$ is the number of points along the passageway, $u_i$ is the gate-perpendicular SIM component, $C_i$ is the SIC at the
$i$th grid cell, and $\Delta x$ is the width of a grid cell (25km).
The corresponding error of SIAF depends on the uncertainties of SIM and SIC products, the sampling number along the
passageways, and the calculation period. For daily SIM vectors, the error was estimated to be about 4.1 km·day$^{-1}$ (Tschudi et
al., 2019). Several assessments indicated an accuracy of about 5% in SIC fields (Peng et al., 2013). Assuming that these two
sources of error are independent, the uncertainty ($\sigma_f$) in estimating SIAF across a 1-km wide gate was estimated at about 2.92,
3.80, and 2.68 km$^2$·day$^{-1}$ for the Fram Strait, S-FJL, and FJL-NZ, respectively. If we assume that the errors of the samples are
additive, unbiased, uncorrelated, and normally distributed, the uncertainty in daily SIAF is $\sigma_D = \sigma_f L / \sqrt{N_s}$ (Kwok, 2009),
where $L$ is the length of the gate, and $N_s$ is the number of independent samples across the gate. From January to June, the
monthly average uncertainties in SIAF through three passageways were estimated to be approximately $1.81 \times 10^3$ to $1.96 \times 10^3$
km$^2$, which were about 3.7%–13.9% of the monthly magnitude and therefore considered acceptable.
To describe the relationship between the SIAF and the sea ice transport before reaching the defined passageway, we also





restructured the sea ice backward drift trajectories from the defined passageways (Fram Strait, S-FJL, and FJL-NZ) over the
three defined periods of January–April, January–May, and January–June 2020, with the ice drifting from the north since 1
January into the passageways by 30 April, 31 May, and 30 June, respectively. The adoption of three periods to restructure the
ice backward drift trajectories is conducive to further distinguishing the difference between the anomalies over the winter
(JFM) or the period of winter through spring (AMJ). In addition, the restructured backward trajectory of sea ice from the
defined passageway can help to identify the source area of the ice, thus reflecting the relationship between the sea ice outflow
and the sea ice conditions in the source area. The sea ice backward drift trajectories were restructured according to Lei et al.
(2019), and the zonal $(X)$ and meridional $(Y)$ coordinates of the backward ice trajectories were calculated as follows:
$X(t) = X(t-1) + U(t-1) \cdot \delta_t$                                                                                              (2)
and $Y(t) = Y(t-1) + V(t-1) \cdot \delta_t$                                                                                       (3)
where $U(t)$ and $V(t)$ are the ice motion components at the time $t$ along the ice trajectories and the $\delta_t$ is the calculation
time step of –1 day. Thereby, the course of time corresponding to the sea ice backward drift trajectory is reversed from the
defined date to 1 January.

In order to reveal the contribution of surface heat budget to sea ice melting, we calculated the potential change in SIT

($\triangle h$) over the time of $\triangle t$, associated with anomalies in surface net heat fluxes over the BGS, according to Parkinson and
Washington (1979):
$-\Delta h = \frac{\Delta t}{\rho L} [\delta FL_{w\downarrow} + \delta FS_{w\downarrow} + \delta H_\downarrow + \delta LE_\downarrow]$                                                                          (4)
where $\rho$ is the density of sea ice (917 kg·m$^{-3}$), $L$ is the latent heat of fusion for sea ice (333.4 kJ·kg$^{-1}$), $\delta FL_{w\downarrow}$, $\delta FS_{w\downarrow}$, $\delta H_\downarrow$
and $\delta LE_\downarrow$ represent the anomalies in surface net fluxes of longwave radiation, shortwave radiation, sensible heat, and latent
heat, respectively, with the positive value denoting the downward heat flux. We note that, the Eq. 4 focuses on the atmosphere-
to-ice heat fluxes but ignores the effects of ocean heat flux. Thus, it can only be used to assess the impact of atmospheric
anomaly on the local sea ice mass balance.
**3. Results**
**3.1 Anomalies in atmospheric circulation patterns**

As shown in Table 1, the monthly AO was in an unusually positive phase from January to March 2020, with the values

maintaining the top three among the years of 1979–2020. And then, the AO decreased to a smaller value in April and turned





to a weakly negative phase in May–June 2020. Monthly CAI in January–June 2020 experienced a continuous positive phase
with an average CAI of 8.5 hPa, which was the largest in 1979–2020. During winter–spring 2020, there were two peaks of
monthly CAI occurring in March and June, with the values of first and fourth in 1979–2020, respectively.

In January–March 2020, accompanied by an unusual positive phase of the AO, the entire Arctic Ocean was almost

dominated by abnormally low SLP compared to the 1979–2020 climatology (the first column of Figure 2). In January 2020, a
large-scale anomalous low SLP appeared near the Kara Sea, and the high-pressure center was observed in Northern North
America. This SLP pattern induced a positive CAI and northerly winds from the high Arctic towards the Barents Sea,
accelerating the southward drift of Arctic sea ice into the Barents Sea and causing regional negative air temperature anomalies
there. In February 2020, the abnormally low SLP dominated near the Barents and Kara Seas, inducing strong northerly winds
in the Atlantic sector of the Arctic Ocean. This SLP pattern continued to cause abnormally high wind speeds over the Atlantic
sector of TPD region, further promoting Arctic sea ice advecting into the BGS and keeping the negative air temperature
anomalies in this region. In March 2020, the low SLP anomalies moved deeper into the central Arctic Ocean and induced
westerly wind anomalies in the BGS.

In April 2020, the low SLP in the Arctic, centered in the northern Beaufort Sea, caused the sea ice to continue to advect

toward the Barents Sea. Subsequently, the SLP structure over the Arctic Ocean has changed greatly in May 2020, with high-
pressure anomalies observed in the Beaufort Sea. The SLP structure in May 2020 was further conducive to Arctic sea ice
advection towards northeastern Greenland. This large change in SLP structure led to the prominently enhanced positive CAI,
which reached the second peak in June, even the AO index decreased remarkably during this period (Table 1). Therefore, the
AO mainly manifests the SLP structure of the pan-Arctic, regulating the sea ice outflow from the TPD region to the BGS by
changing the axis alignment of the TPD. While the CAI mainly affects the wind forcing and ice speed in the TPD region,
especially for the Atlantic sector.

**Table 1.** Monthly AO Index and CAI in winter–spring 2020 and their ranking in 1979–2020



|  | January | February | March | April | May | June |
|---|---|---|---|---|---|---|
| AO | 2.419 | 3.417 | 2.641 | 0.928 | −0.027 | −0.122 |
| Rank | 3rd | 1st | 2nd | 7th | 23th | 26th |
| CAI/ hPa | 4.219 | 11.317 | 19.671 | 5.387 | 2.219 | 7.942 |
| Rank | 11th | 2nd | 1st | 19th | 24th | 4th |



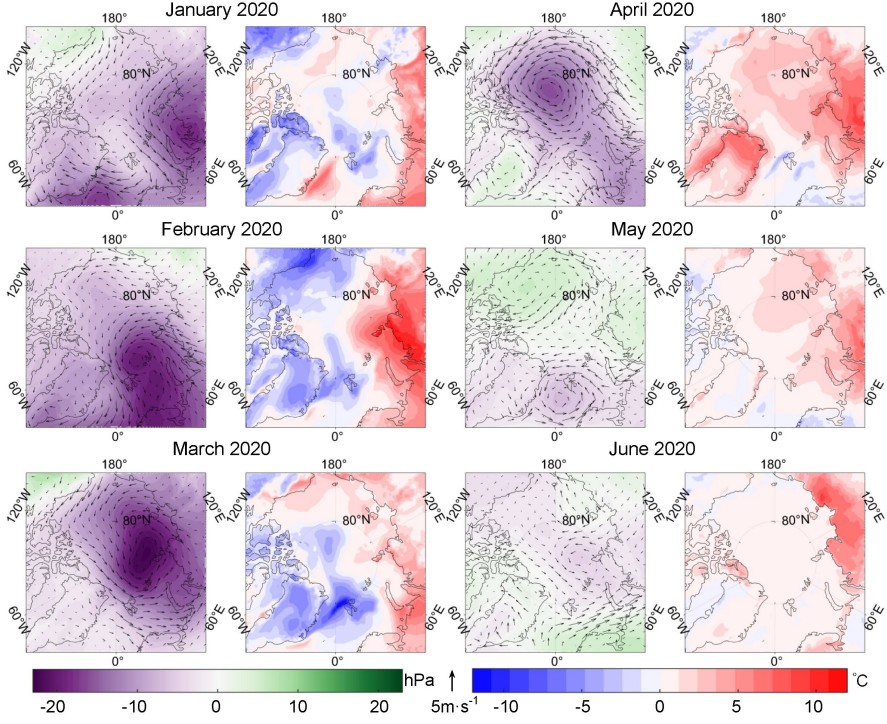


**Figure 2.** Monthly mean SLP (shading) and 10-m surface wind (arrows) anomalies (the first and third columns), and 2-m air temperature

anomalies (the second and fourth columns), during January–June 2020 relative to the 1979–2020 climatology.

**3.2 Anomalies in Arctic sea ice outflow**

We described the SIAF anomalies relative to the 1988–2020 climatology (Figure 3) because differences in satellite data

sources could lead to relatively low SIM speeds derived from the SMMR 37-GHz data during 1979–1987 compared to that

derived from daily SSM/I 85 GHz data, SSMIS 91 GHz and/or AMSR-E 89 GHz observations in the later years (Kwok, 2009).

The cumulative SIAF through Fram Strait and S-FJL both exhibited positive anomalies from January to June 2020. In winter

(JFM) 2020, the cumulative SIAF through the Fram Strait was $1.19\times10^5$ km², which was about 1.2 times the 1988–2020

average, and was the second largest in 2010–2020. Especially in March, the monthly SIAF through the Fram Strait ($5.77\times10^4$

km²) reached the second largest in 1988–2020. The winter cumulative SIAF through the S-FJL in 2020 ($1.51\times10^4$ km²) also

ranked the second largest in 2010–2020. However, the winter cumulative SIAF through the FJL-NZ in 2020 ($2.76\times10^4$ km²)

was only about 81.0% of the 1988–2020 average. This suggests that the sea ice outflow through the FJL-NZ was not sensitive

to the atmospheric circulation pattern of extreme positive AO in winter 2020.

In spring (AMJ) 2020, the cumulative SIAF through the Fram Strait was still at an above-average level, especially with

positive monthly SIAF anomalies in May–June. However, the spring cumulative SIAF through the S-FJL and FJL-NZ was





only 67.5% and 14.1% of the 1988–2020 average, respectively. This implies that the SIAF through these two passageways,
especially for the FJL-NZ passageway in the east, was insensitive to the influence of positive CAI in spring 2020. Consequently,
in January–June 2020, the Fram Strait was the main passageway contributing to the abnormally high total sea ice outflow from
the Arctic Ocean to the BGS, with relative contributions of 73.3% in winter and 86.7% in spring, respectively, responding to
the extreme positive phase of winter AO and the continuous positive phase of winter–spring CAI. In general, in January–
March and June 2020, the accumulated SIAF across three passageways was at the above-average level, with the largest positive
anomalies occurring in March 2020. The abnormally large Arctic sea ice outflow in winter–spring 2020 subsequently
contributed to the dramatic Arctic sea ice loss, resulting in relatively low SIAs of $8.41 \times 10^6$ km$^2$ in June and $5.07 \times 10^6$ km$^2$ in
July 2020, ranking the third and first smallest in 1979–2020, respectively.

The 1988–2020 data has also revealed that the accumulated SIAF through three passageways in both winter and spring

was mainly determined by the SIM speed perpendicular to the passageways ($R = +0.86, +0.85$, respectively; $P < 0.001$). And
in January–February, April and June, the SIM speed in the Atlantic sector of TPD was significantly and positively correlated
with the wind speed (Table A1). Therefore, under the regulation of positive winter AO and winter–spring CAI in 2020, the
relatively high wind speeds led to the larger SIM speeds along the TPD and the increased Arctic sea ice outflow, majorly
through the Fram Strait (Figure 3).

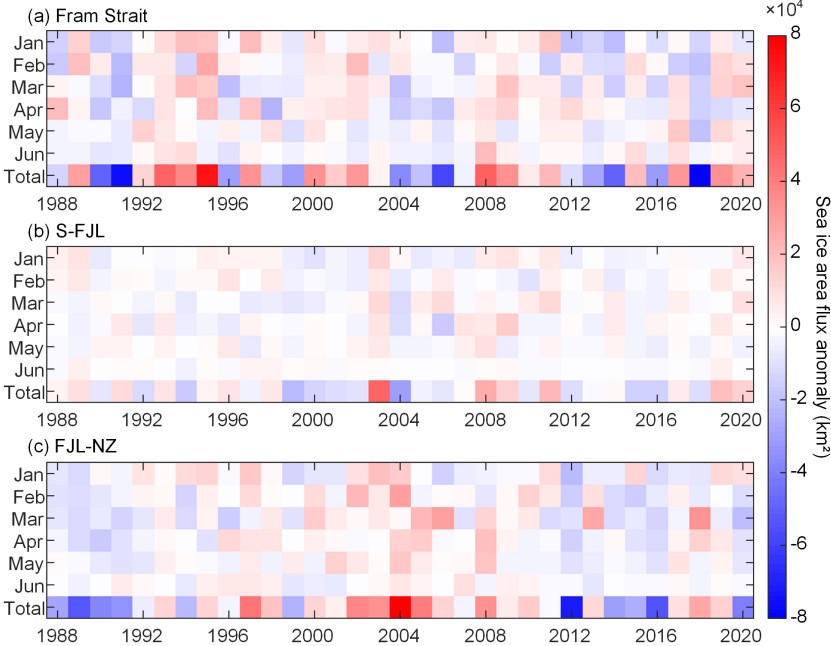


**Figure 3.** Monthly anomalies of sea ice area flux (SIAF) through the Fram Strait, S-FJL, and FJL-NZ from 1988 to 2020. The last row of
each panel represents the anomalies of cumulative SIAF from January to June.





### 3.3 Anomalies in sea ice backward trajectories from the passageways

The backward trajectories can be traced back to the original location of sea ice, thereby denoting the source region of sea ice that advected to the passageways. The broader distribution of the sea ice original area implies that more ice would enter the passageways, leading to an increased sea ice outflow. Compared to the sea ice backward trajectories reconstructed using the average SIM vector of 1988–2020 (Figure 4d–4f), the sea ice backward trajectories from the Fram Strait in 2020 were tilted westward (Figure 4a–4c). This implies that the orientation of TPD was more favorable for exporting thicker ice from the western Arctic Ocean and northern Greenland to the Fram strait during winter–spring 2020. Thus, the anomalies of sea ice volume outflow in winter–spring 2020 were expected more obvious than the SIAF anomalies, if considering that the source region of sea ice was generally dominated by relatively thick sea ice. For the Fram Strait, the endpoints of the sea ice backward trajectories were concentrated at 87°–90°N, which indicates that most of the sea ice advected into this passageway was from the region close to the North Pole. It is worth mentioning that, the restructured sea ice backward trajectory in January–June 2020 was very analogous to that of the MOSAiC ice station (Nicolaus et al., 2021) in the same period, with almost parallel orientation and very close drift distance between them (Figure 4c). Since the slight dislocation was mainly attributed to the inconsistent start point between the reconstructed backward trajectory and the MOSAiC trajectory on 30 June 2020, it increases our confidence in using this method to reconstruct the ice backward trajectories. In all three investigation periods, the net distances from the start points at the defined passageways to the endpoints of the reconstructed ice backward trajectories were the second longest in 1988–2020. In S-FJL, sea ice was mainly advected from the confluence of the Kara Sea and the central Arctic Ocean, and its backward trajectories exhibited a relatively high tortuous feature. However, no reasonable backward trajectories of sea ice could be acquired for the S-FJL passageway according to the temporal starting points of 31 May and 30 June. It was because the relatively low SIC in this region by late spring had restricted the acquisition of valid SIM data. The sea ice advected through the FJL-NZ passageway was mainly from the Kara Sea, which can explain why the change in SIAF through this passageway was insensitive to the changes in the TPD intensity or the CAI pattern.

Overall, compared to the 1988–2020 averages, the sea ice backward trajectories through three defined passageways in winter–spring 2020 were characterized as longer and farther west. Especially, the net distances between the terminal points on 1 January and the starting points from Fram Strait since 30 April, 31 May, and 30 June of each year in 1988–2020 were significantly positively correlated with the corresponding SIAF ($R$ = +0.80, +0.72, +0.75, respectively; $P < 0.001$). Thus, the enhanced sea ice meridional motion along the TPD during January–June 2020 promoted more Arctic sea ice export toward the BGS, which in turn accelerated the reduction of sea ice over the pan Arctic Ocean.


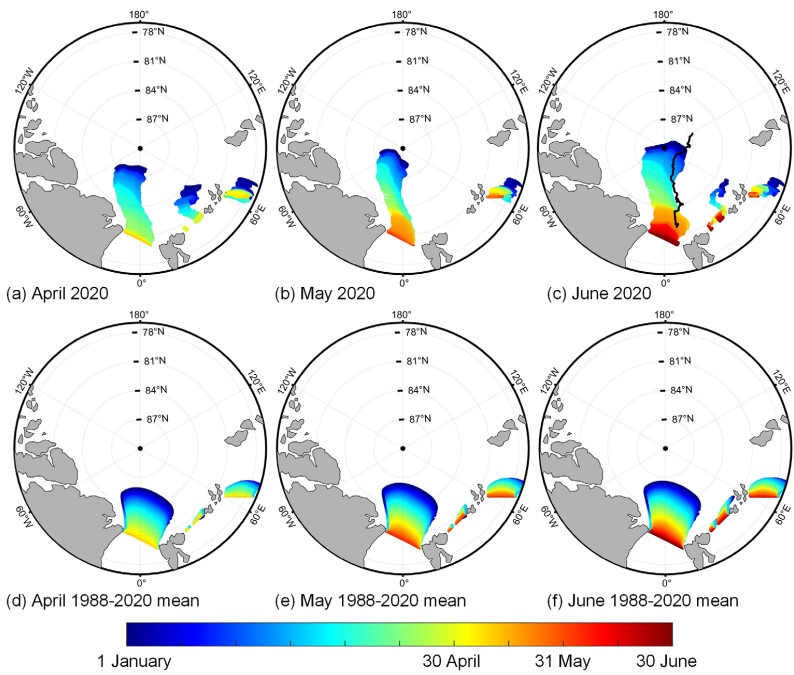

1 January                30 April     31 May     30 June

**Figure 4.** Backward trajectories of sea ice advected to the Fram Strait, S-FJL, and FJL-NZ passageways. The first row shows the backward trajectories of sea ice arriving at the passageways by 30 April, 31 May and 30 June 2020, respectively. The second row is the same as the first row but estimated using the average sea ice motion vector from 1988 to 2020. All endpoints of the reconstructed backward trajectories were set to January 1. The black line in panel (c) represents the MOSAiC trajectories from January 1 to June 30, 2020.

### 3.4 Anomalies in sea ice area and thickness in the Barents and Greenland Seas

SIA in the BGS generally reaches its annual maximum in April each year, since then, as the air and ocean temperature rises, the SIA begins to decrease. In January–May 2020, the SIA anomalies in the Barents Sea are relatively close to 1979–2020 average (Figure 5) and the SIA maintained the top three values in 2010–2020, indicating that the SIA at the study year was less affected by the significant linear decreasing trend. In the Greenland Sea, the SIA anomalies for April–June 2020 are similar to those in the Barents Sea, with the SIA being the first or second largest in 2010–2020. Consequently, in April–June 2020, the SIA in the BGS was much higher compared to the value after removing the linear decreasing trend from 1979 to 2020. Such a large SIA in the BGS during winter–spring 2020 was linked to a more massive sea ice export from the central Arctic Ocean, because a significant relationship ($R = +0.38$, $P < 0.05$) between the anomalies in Arctic sea ice outflow through the three defined passageways and the SIA in the BGS has been identified based on the 1988–2020 data. However, it is worth noting that the impact of sea ice outflow from the Arctic on the SIA in the BGS would be weakened by local processes, such as heat input from the Atlantic water, which reduces the SIA by promoting sea ice melting in the BGS (Lind et al., 2018).



Furthermore, increased sea ice in the BGS, associated with increased subsequent freshwater input to the upper ocean due to
ice melting, was conducive to maintaining oceanic stratification, which in turn constitutes feedback and provides favorable
conditions for the survival of sea ice.

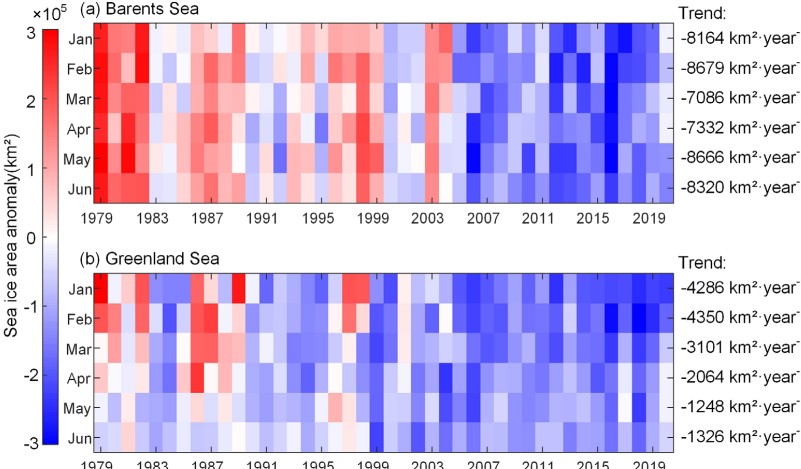


**Figure 5.** Monthly sea ice area (SIA) anomalies in the Barents and Greenland Seas from 1979 to 2020. Also shown on the right are the
corresponding long-term linear trends, which are all statistically significant at the 0.05 level.

As shown in Figure 6, the Greenland Sea initially experienced negative SIT anomalies, and slight positive SIT anomalies

were observed in the Barents Sea during December 2019. Since January 2020, more pronounced positive SIT anomalies were
observed in the Barents Sea and persisted to June, with the most widespread coverage in April–May. This was related to the
anomalous sea ice southward outflow through the S-FJL towards the northern Barents Sea combined with the relatively low
local air temperature. In the Greenland Sea, the SIT anomalies in 2020 turned from negative to positive in March and lasted
until June. This transition also could be attributed to the remarkably increased Arctic sea ice outflow through the Fram Strait,
especially in March 2020.

Since sea ice variability is dominated by both local atmospheric and oceanic forcing (Fery et al., 2015), in addition to sea

ice inflow due to northerly winds, the persistence of negative air temperature anomalies in the BGS from February to April
2020, roughly 2 to 6 °C lower than the 1979–2020 climatology, would also restrict the sea ice melting. Especially in March
2020, negative air temperature anomalies covered almost the entire BGS, and the region with the –6 °C anomalies occurred in
the coincident region with positive monthly SIT anomalies (Figs. 2 and 6). Moreover, compared to the 1979–2020 climatology,
the monthly surface heat fluxes showed upward positive anomalies over the BGS in January–March 2020 (Figure 7), which



were mainly dominated by turbulent heat flux (31.3–40.4 W·m$^{-2}$), accounting for 79.3%–97.1% of the surface heat flux
anomalies. Especially, in February and March 2020, the upward anomalies in sensible heat flux were 1.6–2.2 times larger than
those in latent heat flux. This was likely due to the relatively large air-sea temperature difference and relatively high wind
speed in the BGS during this period, which would result in an unstable atmospheric boundary layer and the increased heat flux
from the ocean to the air (Minnett and Key, 2007). In addition to turbulent heat flux, the net longwave radiation revealed
relatively small upward anomalies (0.9–8.6 W·m$^{-2}$) persisting from January to April 2020, which was also favorable for
preventing ocean warming and ice melting. From April to June 2020, the monthly anomalies in surface heat fluxes were
relatively small, with a value of mostly less than 5 W·m$^{-2}$. It is worth noting that, upward anomalies in net shortwave radiation
were observed in June 2020 over the study region, which coincided with the relatively large SIA and the associated relatively
high regional albedo. The anomalies in cumulative surface heat fluxes from January to June 2020 can be related to a reduced
decrease of 0.01–0.41 m in SIT, estimated using the Eq. 4. Therefore, in general, the heat exchange between atmosphere and
ocean over the BGS, dominated by the upward anomalies in turbulent heat flux in winter 2020, together with the continuous
upward anomalies in net longwave radiation during winter and early spring 2020, as well as the upward anomalies in net
shortwave radiation in June 2020, was conducive to the survival of sea ice during winter and early summer 2020.

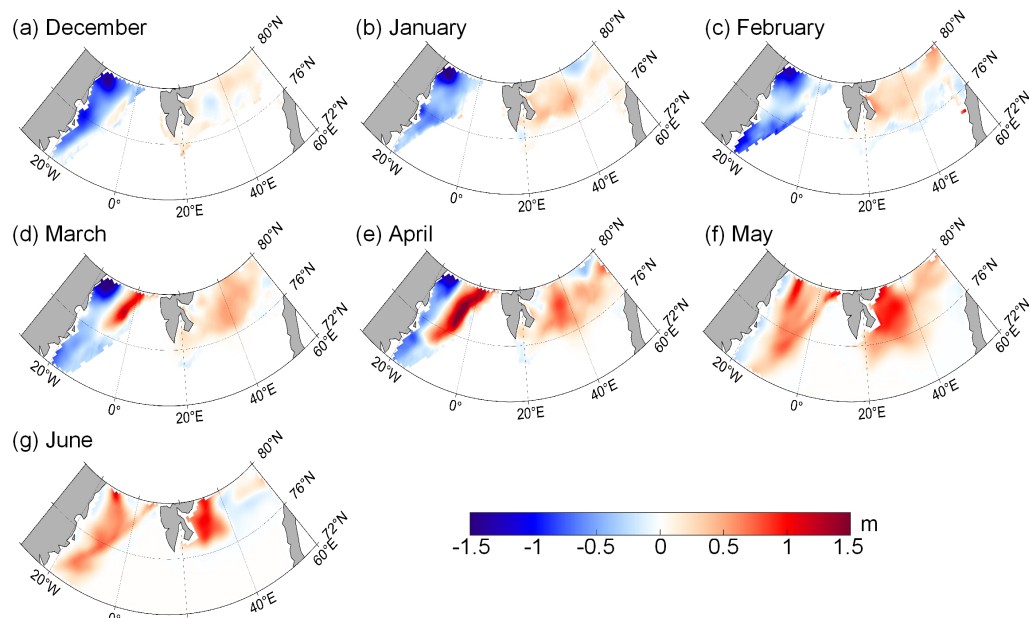


**Figure 6.** Sea ice thickness (SIT) anomalies in the Barents and Greenland Seas from December 2019 to June 2020 compared to the 2011–
2020 average obtained from the CryoSat-2/SMOS product (December–April) and PIOMAS modeled data (May–June).



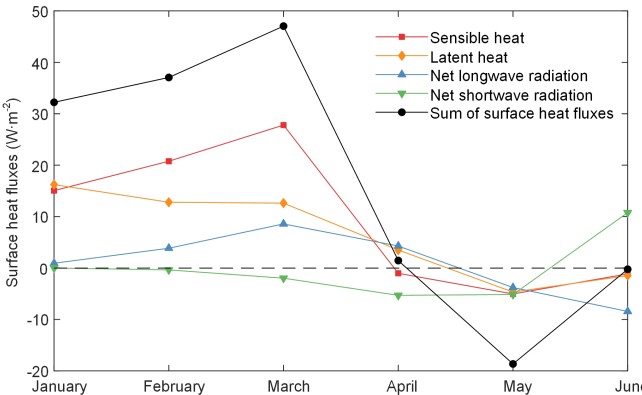

**Figure 7.** Monthly anomalies in surface heat fluxes of sensible heat, latent heat, net longwave radiation, and net shortwave radiation averaged over the study region from January to June 2020 compared to the 1979–2020 average, with positive values denoting the upward fluxes.

## 4. Discussion

### 4.1 Links of Arctic sea ice outflow to atmospheric circulation patterns

In winter–spring 2020, the anomalies in sea ice outflow from the north to the BGS were closely related to the large-scale atmospheric circulation patterns. Furthermore, we quantified the relationship between SIAF and two atmospheric circulation indices (AO and CAI) from 1988 to 2020 to clarify the impact mechanisms of atmospheric circulation on Arctic sea ice outflow. Here, we chose the Fram Strait as the investigated passageway because its SIAF accounts for most (77.6%) of the total SIAF through three passageways. We calculated the correlation coefficient ($R$) between the detrended monthly SIAF and the detrended AO and CAI from January to June for the period 1988–2020 (Table 2). Since the AO is most active in winter, there is a significant positive correlation between SIAF and the AO in February. This is consistent with Rigor et al. (2002), who revealed that more sea ice advection through the Fram Strait was associated with the high positive winter AO. There is also a significant positive correlation between monthly SIAF and CAI in March–April ($P < 0.05$), which suggests that the relatively high CAI could enhance greater southward advection of Arctic sea ice to the BGS, especially during the period (March–April) with a relatively high ice motion speed in the regions north of the BGS compared to other months (Lei et al., 2016).

Furthermore, we examined the years in which extreme high or low (±1 *standard deviation*) phases of the AO or CAI occurred, based on which we obtained the mean SIM field and reconstructed the sea ice backward drift trajectories during January–June in the corresponding years (Figure A1). Associated with the extreme high phase of AO, the sea ice backward trajectories were almost parallel to the prime meridian, i.e., the orientation of TPD was more westward. It means that the positive phase of AO in winter generally leads to a reduction in the spatial scope of Beaufort Gyre and a westward shift of





TPD, which is more conducive to sea ice outflow from the central Arctic Ocean to the BGS (Rigor et al., 2002). Thus, we
believe the relationship between the positive phase anomalies of AO and the westward alignment of TPD identified in 2020,
as shown in Figure 4, is robust. Whereas the sea ice backward trajectories were further to the east under scenarios with the
negative phase of AO. Under the influence of an extremely low AO index, the expanding Beaufort Gyre can weaken the
strength of the TPD and reduces Arctic sea ice export (Zhang et al., 2022). Associated with either the positive or negative phase
of CAI, the sea ice backward trajectories were similar to those under the corresponding phase of the AO. However, in the two
investigated periods of January–May and January–June, there is a higher positive (negative) correlation between the latitude
(longitude) of sea ice backward trajectories endpoints and the CAI compared to the AO (Table A2). This relationship was due
to the fact that the positive phase of CAI might directly enhance the TPD by strengthening wind forcing, hence favoring sea
ice outflow from the central Arctic Ocean into the Fram Strait. The insignificant correlation between them in the investigated
period of January–April may be owing to the fact that the sea ice backward trajectories restructured in this period were
relatively short range and the variations in the locations of the backward trajectory endpoints between the years were relatively
small.
**Table 2.** Correlation coefficient ($R$) between monthly sea ice area flux (SIAF) through the Fram Strait and atmospheric
circulation indices in 1988–2020

| Month | January | February | March | April | May | June |
|---|---|---|---|---|---|---|
| AO | n.s. | 0.437* | n.s. | n.s. | n.s. | n.s. |
| CAI | 0.610*** | n.s. | 0.403* | 0.538** | n.s. | n.s. |

Note: Significance levels are $P < 0.001$ (***), $P < 0.01$ (**) and $P < 0.05$ (*); n.s. denotes nonsignificance at the 0.05 level.
**4.2 Impact of sea ice anomalies on the hydrographical and ecological conditions in the Barents and Greenland Seas**
In April–June 2020, the BGS experienced widespread negative anomalies of SST (–1°C to –3 °C), with monthly SSTs being
the lowest in 2005–2020 (Figure 8). Furthermore, the small negative SST anomalies over the Barents Sea persisted to August
2020. The detrended correlations between the monthly SIA and contemporaneous SST in the BGS from April to June over
1982–2020 (Table A3) were significantly negative. The impact and feedback mechanisms can be summarized as that the
abnormally large Arctic sea ice outflow in winter–spring 2020 led to an increased SIA and the associated relatively high albedo
in the BGS, thereby preventing the absorption of incoming solar radiation by the ocean and suppressing the rise in SST. In
turn, relatively colder seawater was not conducive to sea ice melting there. The corresponding correlation coefficients in the
Greenland Sea were weaker compared to those in the Barents Sea, which may be due to the relatively complex influence
factors on the SST variations in the Greenland Sea. That is to say, the northwestern Greenland Sea is suppressed from cooling
effects due to sea ice and surface current inflow from the north, while the southeastern part is subject to warming effects from





warm Atlantic heat flow (Wang et al., 2019). Regionally, we found that the negative correlation coefficients between SIA and
SST are larger in the southern BGS (76°–80°N) than in the northern part (72°–76°N). This is likely because the SST is more
closely correlated with the SIC in areas with less sea ice (Wang et al., 2019). In addition, we examined the statistical
relationship between the April SIA and the monthly SST with a lag of 1–3 months in the BGS (Table A4). It manifests that
there was a significant negative correlation between them with a lag of 1–2 months, with the decreased correlation coefficients
as the increased lagging time. In the Barents Sea, the April SIA still had a significant negative effect on the increase in SST
until July, i.e., with a lag of 3 months, whereas in the Greenland Sea, the significant influence of April SIA on the SST only
lasted until June. This difference suggests that the sea ice anomalies in the Barents Sea have a longer memory for the impact
on the SST than those in the Greenland Sea.

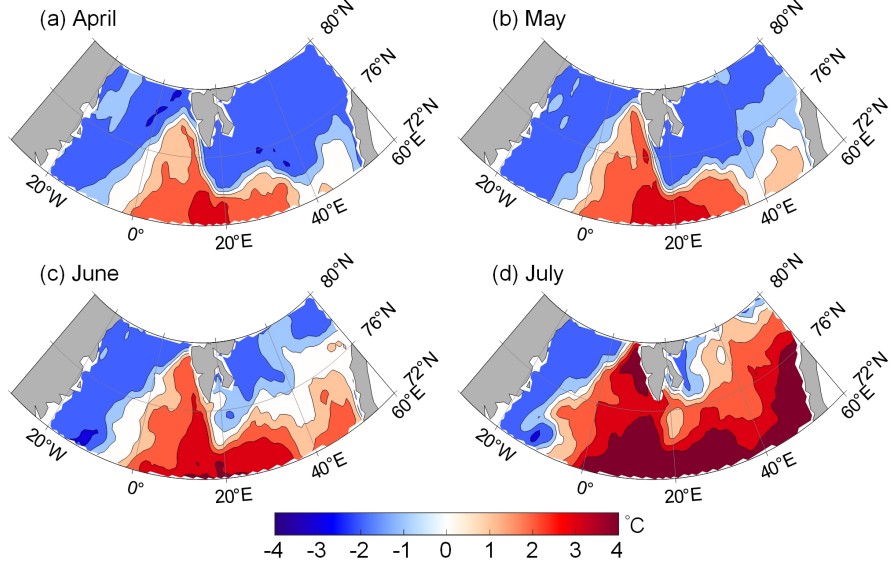


**Figure 8.** Monthly sea surface temperature (SST) anomalies in the Barents and Greenland Seas from April to July 2020 compared to the
2005–2020 average.
Arctic marine primary producers using photosynthetic light during spring bloom are largely restricted by sea ice cover
(Campbell et al., 2015). The *Chl-a* over the southern Greenland Sea in April 2020 was smaller compared to the previous 5
years. A significant negative correlation between *Chl-a* and SIA in April over 1998–2020 was identified ($R = -0.45$, $P < 0.05$).
This implies that the increase in SIA inhibited the growth of marine primary producers, as the sea ice reflected most of the
solar shortwave radiation back to space and was therefore not favorable for the growth of phytoplankton in early spring. The
relatively low *Chl-a* in April 2020 was accompanied by the occurrence of abnormally low SST. In general, the relatively low
SST is detrimental to the melting of sea ice, which reduces the absorption of radiation by the upper ocean and weakens



photosynthetic activity (Brown et al., 2011). However, there is no significant correlation between *Chl-a* and SST in the BGS.
This may be due to the complex interactions between SST, SIC and *Chl-a*, which together affect the changes in *Chl-a* (Arrigo
and van Dijken, 2015; Siswanto, 2020). And the effect of a single SST on *Chl-a* may be limited. Compared to the 2005–2020
average, *Chl-a* in 2020 started to reveal positive anomalies in May and persisted to June (Figure 9). This implies that the
conditions in later spring 2020 were well suited for the growth of marine primary producers in the BGS. It was likely because
1) the high ice coverage in early spring was conducive to phytoplankton seeding, and 2) the low primary producers in early
spring were beneficial to the residue of marine nutrients. Seasonally, the *Chl-a* in the BGS reached its peak in May–June of
the year, one month later than the peak of SIA, which can be considered normal compared to previous observations (e.g.,
Dalpadado et al., 2020; Siswanto, 2020). Thereby, the impact of the abnormally large SIA in winter 2020 on spring *Chl-a* was
mainly limited to April 2020. Thus, the abnormal Arctic sea ice flow plays an identifiable role in regulating the seasonal timing
of the BGS ecosystem.

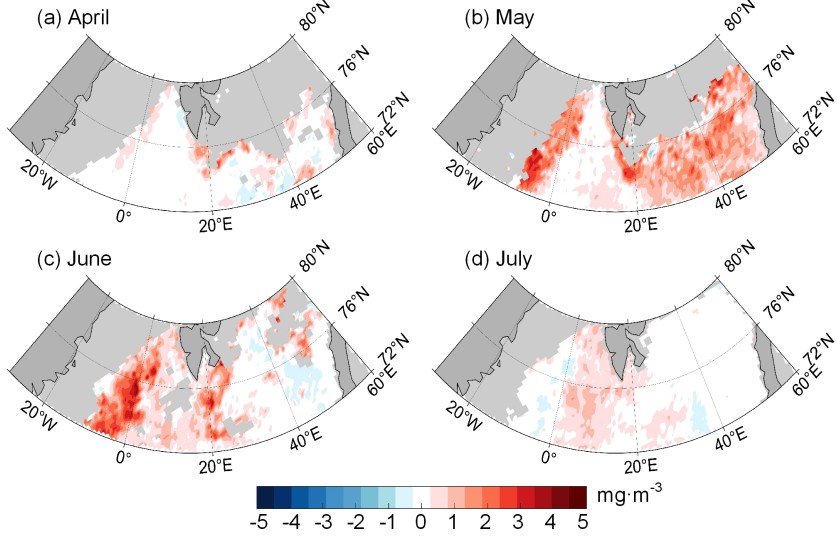


**Figure 9.** Monthly *Chlorophyll-a* (*Chl-a*) anomalies in the Barents and Greenland Seas from April to July 2020 compared to 2005–2020
average.
**4.3 Are the anomalies and their connections identified in winter–spring 2020 typical in climatology?**
In the past decade, positive anomalies in the winter–spring SIAF through the three defined passageways relative to the
1988–2020 climatology were also identified in 2011, 2017, and 2019, close to the value in 2020 (Figure 3). Therefore, we also
quantified the anomalies of sea ice and ocean conditions in the BGS for these years, so as to assess the representativeness of
the seasonal feedback mechanisms identified in winter–spring 2020 under the scenario of abnormally high Arctic sea ice



outflow. During these three years, the sea ice backward trajectories reconstructed starting since 30 April, 31 May, and 30 June
were also characterized as longer and farther west compared to 1988–2020 climatology. This suggests that the ice speeds along
the TPD were relatively large and could partially contribute to the positive SIAF anomalies in these years. In the BGS, although
small negative SIA anomalies were observed in March–June 2011, 2017, and 2019 compared to the 1979–2020 climatology,
their values were still much higher than those estimated from the long-term linear decreasing trends since 1979 by $0.16 \times 10^4$–
$2.79 \times 10^4$, $0.43 \times 10^5$–$1.38 \times 10^5$ and $0.66 \times 10^5$–$1.06 \times 10^5$ km$^2$, respectively. During these three years, similar upward anomalies
in accumulated net surface heat fluxes were also identified in January–March, suggesting the potential coupling mechanism
between sea ice coverage and surface heat budget in the BGS. However, compared to the 1979–2020 climatology, there were
positive air temperature anomalies in January–March 2011, 2017, and 2019, in contrast to the negative air temperature
anomalies in 2020. This may subsequently contribute to the relatively small negative SIA anomalies in these years than in
2020. The SIT anomalies were calculated only for 2017 and 2019 since satellite SIT data were not available prior to 2011, and
we found that the BGS also showed small positive anomalies from March to June for both years compared to the average since
2011. Furthermore, the sea ice anomalies in these years also had impacts on the marine hydrographical and ecological
conditions of the BGS in April–June. The monthly SSTs in May–June of 2011, 2017, and 2019 all maintained the 2nd–4th
lowest in 2010–2020. During these years, the *Chl-a* also showed relatively pronounced negative anomalies in April. By
comparing with these years that also experienced abnormally large Arctic sea ice outflow, it can be considered that the sea ice
anomalies and their connections to the marine environments in the BGS identified in winter–spring 2020 were representative.
However, we also expect that the influences of abnormally high Arctic sea ice outflow on the sea ice and other marine
conditions in the BGS will gradually weaken if the Arctic sea ice continues to thin and the northward Atlantic Ocean heat flow
continues to increase, because the thinner ice under the increased oceanic heat would not be conducive to the survival of sea
ice in the BGS.
**5. Conclusions and recommendations**
In this study, we investigated the impacts of the anomalies of atmospheric circulation and Arctic sea ice outflow in the
winter and spring of 2020 on the sea ice conditions in the TPD downstream region of the BGS, and then discussed the
connections between winter–spring sea ice anomalies and the hydrographical and ecological conditions of the BGS in the
subsequent months until early summer 2020.
Compared to the 1979–2020 climatology, the AO experienced an unusually large positive phase in January–March 2020.
In the context of this, the SLP structure, associated with the positive CAI induced strong northerly winds along the Atlantic



section of TPD, which then facilitated Arctic sea ice outflow to the BGS. In the following three months, the AO decayed to be
negative, while the CAI remained positive, which ensured a continuous enhanced Arctic sea ice outflow to the BGS. Therefore,
in January–March and June 2020, the total SIAF through three passageways north of the BGS was extremely large compared
to the 1988–2020 climatology, mainly through the Fram Strait, which accounts for 77.6% of the total SIAF. The variabilities
of seasonal accumulated SIAF in 1988–2020 through these passageways were mainly dominated by the change in SIM ($R =$
$+0.86$ for January–June; $P < 0.001$), and it was significantly positively correlated with AO in February, and with CAI in March
and April ($P < 0.05$). Under the positive phases of AO and CAI in winter and/or spring 2020, the sea ice backward trajectories
reaching Fram Strait were relatively longer and sloped westward compared to the 1988–2020 climatology, which reflects the
larger ice speed along the TPD and the orientation of the TPD favoring Arctic sea ice outflow to the BGS. This regime also
manifests that AO affects Arctic sea ice outflow by modifying the axis alignment of TPD, while the CAI directly affects the
wind forcing in the TPD region.
The abnormally high sea ice outflow through the Fram Strait and S-FJL in winter–spring 2020 subsequently affected the
SIA and SIT in the BGS in the spring and early summer of 2020. In addition, the regional low air temperature anomalies in the
BGS favored the survival of sea ice there. Furthermore, relatively large upward anomalies in surface heat fluxes dominated by
turbulent heat flux in winter 2020, continuous upward anomalies in net longwave radiation in winter and early spring 2020,
and upward anomalies in net shortwave radiation in later spring 2020 can also reduce ice melting in the BGS. Thus, the monthly
SIA in the BGS in April–May 2020 remained the first or second largest in 2010–2020, and the relatively large SIT over the
BGS was observed since March 2020, especially in May–June. Furthermore, sea ice anomalies in the BGS subsequently
influenced the hydrographical and ecological conditions in the spring and early summer of 2020. In this region, the SIA in
April was significantly negatively correlated with the synchronous SST, as well as that with a lag of 1–3 months. And the SST
in April–June 2020 was the lowest in 2005–2020. In the Greenland Sea, there was a significant negative correlation between
the April SIA and synchronous *Chl-a*, which implies that high SIA could weaken photosynthetic activity and inhibit
phytoplankton blooms in early spring. A comparison with similar scenarios with a high Arctic sea ice outflow in other years in
the recent decade confirmed that the relationships between sea ice anomalies and the hydrographical and ecological conditions
in the BGS identified in winter–spring 2020 is representative. This suggests that the winter–spring Arctic sea ice outflow could
be considered a predictor of the changes in the conditions of sea ice and other marine environments in the BGS in the
subsequent months, at least until early summer.
In this study, we used remote sensing retrieval products of SST and *Chl-a* to characterize the apparent hydrographical and
ecological status in the BGS, which is very insufficient for a thorough understanding of the dynamical coupling mechanism of





sea ice, ocean, and biology. Remote sensing data can only reflect seasonal variations in net primary productivity in ice-free
oceans, whereas changes in primary productivity of ice algae and ice-submerged phytoplankton ecosystems are still not
quantifiable. Thus, it would be recommended to further collect the in situ observation data of regional physical oceanography,
biology, and ecology, as well as biogeochemical cycles to characterize the impact mechanisms of the abnormal Arctic sea ice
outflow on the oceanic, ecological, and biogeochemical processes in the study region. In particular, how the seasonal evolutions
of ocean stratification, mixing and frontal dynamics, biological communities, and greenhouse gas fluxes between ocean and
atmosphere respond to and/or feedback to the changes in sea ice is a scientific focus worth of attention, associated with the
increased Arctic sea ice outflow into the BGS region. Further studies will build on the results presented here.
**Appendix A: Extra figures and tables**

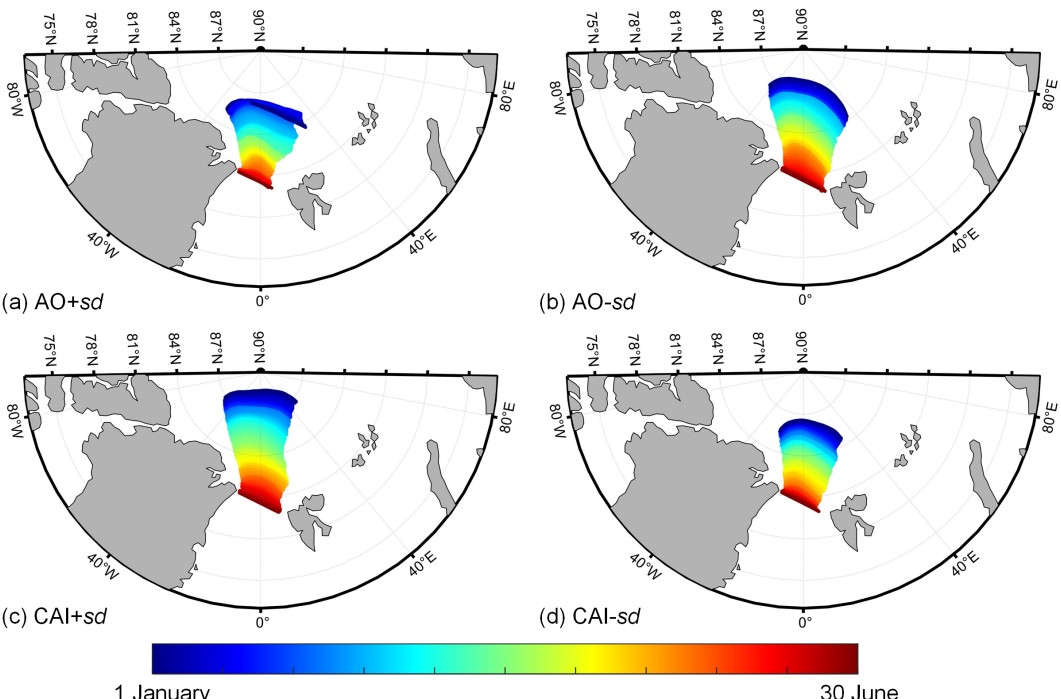


**Figure A1.** Sea ice backward trajectories from the Fram Strait under the extreme positive and negative phases of the Arctic Oscillation (AO)
and Central Arctic west-east air pressure gradient Index (CAI) in 1988–2020. Color coding of the sea ice backward trajectories denotes the
time from 1 January to 30 June.





**Table A1.** Correlation coefficient ($R$) between monthly sea ice motion speed and wind speed in the Atlantic sector of TPD for 1979–2020

| Month | January | February | March | April | May | June |
|---|---|---|---|---|---|---|
| $R$ | 0.411** | 0.355* | n.s. | 0.478** | n.s. | 0.493*** |

Note: Significance levels are $P < 0.001$ (***), $P < 0.01$ (**) and $P < 0.05$ (*); n.s. denotes nonsignificance at the 0.05 level.

**Table A2.** Correlation coefficient ($R$) between the latitude or longitude of sea ice backward trajectory endpoint from the Fram Strait and atmospheric circulation indices in 1988–2020

| Investigation period | January–April | January–May | January–June |
|---|---|---|---|
| Lat *vs.* AO | n.s. | 0.354* | 0.347* |
| Lon *vs.* AO | n.s. | −0.419* | −0.514** |
| Lat *vs.* CAI | n.s. | 0.625*** | 0.590*** |
| Lon *vs.* CAI | n.s. | −0.508** | −0.599*** |

Note: Significance levels are $P < 0.001$ (***), $P < 0.01$ (**) and $P < 0.05$ (*); n.s. denotes nonsignificance at the 0.05 level.

**Table A3.** Synchronous correlation coefficient ($R$) between monthly sea ice area (SIA) and sea surface temperature (SST) in April, May, or June for 1982–2020.

| | Month | All | North(76°–80°N) | South(72°–76°N) |
|---|---|---|---|---|
| | April | −0.917*** | −0.764*** | −0.916*** |
| Barents Sea | May | −0.836*** | −0.706*** | −0.810*** |
| | June | −0.750*** | −0.677*** | −0.704*** |
| | April | −0.640*** | n.s. | −0.394* |
| Greenland Sea | May | −0.661*** | n.s. | −0.409** |
| | June | −0.656*** | n.s. | n.s. |

Note: Significance levels are $P < 0.001$ (***), $P < 0.01$ (**) and $P < 0.05$ (*); n.s. denotes nonsignificance at the 0.05 level.



**Table A4**. Lagging correlation coefficient ($R$) between monthly sea ice area (SIA) in April and sea surface temperature (SST) in May, June, or July for 1982–2020.

|  | Month | All | North(76°–80°N) | South(72°–76°N) |
|---|---|---|---|---|
| Barents Sea | May | −0.851*** | −0.651*** | −0.874*** |
|  | June | −0.752*** | −0.623*** | −0.739*** |
|  | July | −0.459** | −0.529*** | −0.364* |
| Greenland Sea | May | −0.564*** | n.s. | n.s. |
|  | June | −0.446** | n.s. | n.s. |
|  | July | n.s. | n.s. | n.s. |

Note: Significance levels are $P < 0.001$ (***), $P < 0.01$ (**) and $P < 0.05$ (*); n.s. denotes nonsignificance at the 0.05 level.

**Data Availability**

Sea ice motion data from the NSIDC is available at https://nsidc.org/data/NSIDC-0116/versions/4 (last access on 31 Dec 2021). NSIDC sea ice concentration data is obtained from https://nsidc.org/data/G02202/versions/4 (last access on 31 Dec 2021). Sea ice area data in the Northern Hemisphere is available at https://nsidc.org/data/g02135/versions/3 (last access on Oct 2022). Sea ice thickness is downloaded from merged CryoSat-2 and SMOS (https://data.seaiceportal.de/data/cs2smos_awi/v204/; last access on 10 Apr 2022) and PIOMAS (https://pscfiles.apl.uw.edu/zhang/PIOMAS/; last access on 31 Dec 2020). Sea surface temperature data is available at https://psl.noaa.gov/data/gridded/data.noaa.oisst.v2.highres.html (last access on real-time). *Chl*-a data is obtained from https://climate.esa.int/en/projects/ocean-colour/data/ (last access on Dec 2021). The ERA5 atmospheric reanalysis data are downloaded from https://cds.climate.copernicus.eu/cdsapp#!/dataset/reanalysis-era5-single-levels (last access on real-time). The AO index is available at https://www.cpc.ncep.noaa.gov/products/precip/CWlink/daily_ao_index/ao.shtml (last access on Oct 2022).

**Author Contributions**

FZ carried out the analysis, processed the data, and prepared the manuscript. RL provided the concept, discussed the results, and revised the manuscript during the writing process. All authors commented on the manuscript and finalized this paper.



**Competing Interests**
The authors declare that the research was conducted in the absence of any commercial or financial relationships that could be
construed as a potential conflict of interest.
**Financial support**
This work was financially supported by the National Key Research and Development Program (grant nos. 2021YFC2803304
and 2018YFA0605903) and the National Natural Science Foundation of China (grant nos. 41976219 and 42106231).

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
