# Peer review of "The impacts of anomalies in atmospheric circulations on Arctic sea ice outflow and sea ice"

_The Cryosphere, 2022_

## Author Comment (AC1)

**Response to RC1**

Thank you for your time and constructive comments on the manuscript "Impacts of anomalies in Arctic sea ice outflow on sea ice in the Barents and Greenland Seas during the winter-to-summer seasons of 2020". We would consider each comment carefully and incorporate practically all of them in the revised manuscript.

**Major comments**

1Title:

I would like to see a better title. The current title read: "Impacts of anomalies in Arctic sea ice outflow on sea ice in the Barents and Greenland Seas during the winter-to-summer seasons of 2020" When I look the following text, I felt the manuscript is mainly dealing with the factor that create the Arctic sea ice outflow anomalies, as author stated in the abstract (L9): "the impacts and feedback mechanisms on a seasonal scale of anomalies" So, I suggest authors to speak out what "impacts", e.g. atmospheric circulations. One possibility could be: The impact of atmospheric circulations on the anomalies of sea ice outflow and their feedback mechanisms in the Barents Sea and Greenland Sea

Reply: Thank you for comments. We will revise the title to "The impact of atmospheric circulations on the anomalies of sea ice outflow and their feedback mechanisms in the Barents Sea and Greenland Sea" following this suggestion, to emphasize the impacts of atmospheric circulation on sea ice outflow.

2 Abstract:

There are totally 322 words. I think it is too long, please compact it to e.g., 250 words. However, if TC accepts a long abstract, so be it, but please add some compact analyses/statement to echo latest state of the art findings. I am sure there are papers dealing with Arctic sea ice outflow

Reply: Thank you for pointing this out. We will abstract and add some key analysis results from our study.

3 Introduction:

This part is largely ok, but as I have stated in previous point, please consider echoing your work with UpToDate finding. I recommend authors to check Sumata et al., 2022. This paper should be cited in your work. Some comparison would be even better in results/discussion section. The language can still be improved. This comment valid for entire manuscript.

Reply: In the revised manuscript, we will quote some latest literature to echo our work, especially Sumata et al. (2022). Through comparing the results derived from this paper, it is conducive to further enriching our research conclusions. In section 4.3, we will add some analysis to compare the scenario with the relatively low Arctic sea ice outflow discussed by Sumata et al. (2022). We will further improve the expression and language throughout the manuscript.

4. Data and method:
I am quite impressed that such comprehensive data sets are used in this work, well done. I wish authors could make further elaborate on data accuracy and comment/assess the data consistency, for example, authors wrote "Here, we used the CryoSat- 122 2/SMOS SIT from December to April, and the PIOMAS SIT from May to June in 2011–2020 to estimate the anomaly in SIT during the study year of 2020". Do I need to worry about the inconsistency of the data sets applied here?
Reply: According to this comment, we will further elaborate on the accuracy of the data and assess its consistency, especially for the sea ice thickness products we used.

Line: 180-182: "We note,,,". So, this do suggest that in study deals with the impact of the atmosphere on sea ice not ocean at all. I suggest authors express this argument explicitly already in the beginning, e.g., introduction.
Reply: In the revised paper, we will follow this suggestion and state in the introduction what our study focuses on.

5 Results:
Figure 2 is very comprehensive and informative, yet in the main text, I see only once Figure2 (the first column of Figure 2), please add more instruction on what text explain/analyses other columns of the Figure 2.
Figure 3 is also very informative; I suggest you separate last row of each panel to make it clearer and easier to distinguish from others. Furthermore, any patterns can be extracted from this figure?
3.3 section is very interesting. However, in order to prove the effectiveness of the reconstructed results of the ice floe backward drift trajectory, it would be interesting to compare the ice floe backward drift trajectory with forwarded observed buoy drift trajectories for example, under the scenarios of AO+, AO-, CAI+, CAI – and see whether the buoys' drift trajectories are consistent with the reconstructed results. If not, any impact on your results and conclusions

Could you elaborate further whether or not the abnormal AO and CAI would have impacts on sea ice thermodynamics, e.g. total ice mass balance before ice floes reached the Fram Strait? I would like to see more discussions.

Reply: Thank you for the constructive comment, the following is the corresponding response:

1)  In fact, section 3.1 contains some instructions for Figure 2, but does not indicate the corresponding columns of the figure. We will add the corresponding column indications of Figure 2 after the corresponding text.

2)  We will separate the last row of each panel of Figure 3 to make the image clearer and easier to understand. We will add some summary text extracted from Figure 3 in section 3.2 to highlight the knowledge obtained from Figure 3.

3)  We will add some analysis in Section 3.3 to verify the validity of the reconstruction results of sea ice backward drift trajectory. We plan to use the observed buoy data for validation. For the selection of buoy data, we choose buoys with drift trajectories in January–June during the years with the AO+, AO-, CAI+, and CAI- atmospheric circulation patterns.

4)  To further check whether there are some influences of the atmospheric circulation pattern with abnormal AO and CAI on sea ice thermodynamic process, we will add the corresponding discussions. First, we calculate the average trajectory of the sea ice backward trajectory for the AO+, AO-, CAI+, and CAI- cases during January–June. After obtaining the reconstructed trajectories, we will compare the Freezing Degree Days (FDD)—the integral of air temperature below the freezing point over the freezing season, and the reanalysis data of atmospheric surface heat flux obtained along the reconstructed trajectories in the years with different atmospheric circulation patterns to assess the influence of AO and CAI on the changes in sea ice thermodynamic process. In addition, we note that Sumata et al., (2022) found that the sea ice in the region south of 82°N near the Fram Strait was affected by strong heat supply from the ocean and thus melted rapidly. Therefore, we will compare the lengths of time that the reconstructed trajectory within the region south of 82°N before the floe reached the Fram strait in the case of abnormal AO and CAI to further illustrate the effect of atmospheric circulation patterns on the thermodynamic mechanism of sea ice.

6 Discussion:
Please strengthen the linkages of your work with the latest state of the art of research e.g., Sumata et al., (2022). In such extreme season, what are the possible impacts of Arctic sea

ice outflow on the a) sea ice state and b) marine hydrographical and c) ecological conditions in the Barents Sea and Greenland Sea?

Reply: In order to strengthen our links with the latest research, we will add some comparative discussions in section 4.3. We selected 2018, mentioned in Sumata et al. (2022), to obtain the effects of abnormally low Arctic sea ice outflow on sea ice state, marine hydrographical and ecological conditions in the Barents and Greenland Seas. The results will be compared with that given by Sumata et al. (2022).

7 Conclusions and recommendations

I suggest you drop recommendations because you merely "recommended to further collect the in situ observation,,,, in the study region" which is not necessarily entitled as recommendations, unless if you recommend some specific concrete parameters/variables or some specific instrumentation to be observed or to be used and further to be linked to each other.

Reply: We will drop the recommendations and revise the section title as the suggestion.

**Minor comments**

2) Figure 1: There is no need to define a geometrically regular study regions for the BGS. Its northern boundary can be consistent with the defined passageways, and the area bordering Greenland and other islands can be consistent with the shoreline.

Reply: Good point. We will change the defined study area of the Barents and Greenland Seas in Figure 1 as the suggestion.

3) Line 107 "In addition, we used data from the NSIDC Sea Ice Index version 3 (Fetterer et al., 2017) to obtain monthly SIA changes in the Northern Hemisphere in 2020." The purpose of using this data is unclear. In addition, data from the Arctic should be used instead of data from the Northern Hemisphere.

Reply: We will correct this description mistake of the data and add the purpose of using the data in the revised manuscript.

14) Section 4.3, 1) also consider the scenario with low Arctic sea ice outflow, 2) Does North Atlantic Oscillation have a significant regulatory effect on the marine environment of BGS?

Reply: We will add some discussions on the impact and feedback under the scenario of abnormally low Arctic sea ice outflow. We will also further test the synchronous and

lagging correlation between the North Atlantic Oscillation and sea ice and oceanographic parameters in the BGS region.

1) Line 50: "plays a crucial role in shaping the icescape in this region"-- change the "shaping" to "proving the preconditions"

4) Line 205 "regulating the sea ice outflow from the TPD region to the BGS" change to "regulating the sea ice outflow from the Arctic Ocean to the BGS"

5) Line 234 "resulting in relatively low SIAs of" change to related to..., Arctic sea ice outflow is only one of the factors affecting the reduction of Arctic sea ice.

6) Line 266 "was insensitive to the changes in the TPD intensity or the CAI pattern" delete "the TPD intensity or" because you did not directly quantify the strength of TPD.

7) Line 309 "the monthly surface heat fluxes" (and also other text) change to "the monthly atmospheric surface heat fluxes"

8) Line 323 "sea ice during winter and early summer 2020." change to "sea ice during spring and early summer 2020."

9) Line 370 "the absorption of incoming solar radiation" delete the "incoming".

10) Line 376 "are larger in the southern BGS (76°–80°N) than in the northern part (72°–76°N)"-- this should be a mistake.

11) Line 395 "the complex interactions between SST, SIC and Chl-a" change to "the complex interactions between SST and SIC".

12) Line 401 "the year" change to "the study year".

13) Line 403 "the abnormal Arctic sea ice flow" change to "the abnormal Arctic sea ice outflow".

15)Tables in Appendix: consider using simple expressions to indicate different significant levels, e.g., text in bold, or Italic.

It would be nice to apply professional language service for the entire manuscript

Reply: Thank you for your careful advice. All grammatical mistakes and inappropriate expressions will be revised as suggestions. We will apply for professional language services to improve the language throughout the entire manuscript.

---

## Author Comment (AC2)

**Response to RC2**

Thank you for your time and constructive comments on the manuscript "Impacts of anomalies in Arctic sea ice outflow on sea ice in the Barents and Greenland Seas during the winter-to-summer seasons of 2020". We would carefully consider each comment, and make corresponding changes based on the feedback provided in all cases.

**Minor comments:**

Lines 19–20: I think "an extremely low Chlorophyll-a concentration observed over the BGS in April" is not supported by Figure 9. The anomaly of Chlorophyll-a concentration over the BGS in April 2020 is small shown by Figure 9a.

Reply: Thank you for comments. We will revise this sentence. In fact, the extremely low Chlorophyll-a concentrations only occur over the southern Greenland Sea in April 2020.

Line 47: Change "in the BGS" to "in the Barents Sea".

Reply: We will revise the inappropriate description.

Figure 2: Add ice drift velocity anomalies to the second and fourth columns.

Reply: Thank you for the reminder. We will calculate corresponding anomalies of sea ice drift velocity and add them to the second and fourth columns of Figure 2.

Lines 251–253: "the anomalies of sea ice volume outflow in winter–spring 2020 were expected more obvious than the SIAF anomalies" may be not true. Because the ice thickness has larger negative trend than ice concentration during winter–spring over these regions, so it may lead to the volume outflow anomaly ranking is less obvious than the SIAF anomalies.

Reply: We realize that such a sentence is not as rigorous, although we assume that most of these outflows are relatively thicker ice to reach such a conclusion. Since we lack observations of sea ice thickness in the outflow regions to quantify the sea ice volume outflow anomaly, we will revise this supposition.

Lines 319–320: How to estimate SIT changes using the Eq.4 is not very clear. I think the surface heat flux anomalies shown in Figure 7 are the results for the whole BGS. Considering parts of BGS are ice-free, it is not a good idea to estimate SIT using the anomalies over the whole BGS. Using the surface heat flux anomalies only over the climatological ice-covered regions is more reasonable.

Reply: This is a good point. Using surface heat flux anomalies in the entire BGS to estimate SIT changes is really not reasonable. We will re-estimate the change in SIT using Equation 4, using the surface heat flux anomaly over the climatological icecovered regions of the BGS.

Reply: According to the definition of extreme high (low) phase of AO and CAI, there are 2 (5) years of extreme positive (negative) for AO and 6 (4) years of extreme positive (negative) for CAI in the period 1988–2020. The number of extreme high (low) years will be given in appendix Figure A1 for illustration purposes.

Reply: Following this suggestion, we will add a figure in the appendix, which is a time series plot of regional mean Chl-a concentration, to show the changes in Chl-a over the southern Greenland Sea in April 2005–2020.

---

## Author Comment (AC3)

**Response to RC3**

We truly appreciate the reviewer for the careful reading and constructive comments on the manuscript "Impacts of anomalies in Arctic sea ice outflow on sea ice in the Barents and Greenland Seas during the winter-to-summer seasons of 2020". We would thoroughly consider all comments with meticulous consideration, and endeavor to incorporate all of them into the revised manuscript.

The followings are our preliminary responses to these comments:

Unfortunately, the manuscript as a whole lacks for clarity in my view. I have read this manuscript several times, and I am still not sure I can summarise the main findings. There is a lot of interesting information in the paper, but I find it hard to follow the *reasoning* - it was often not clear to me what one should make of the specific results that were presented, and I often had a hard time understanding on what basis the authors arrived at their conclusions. The discussion section currently consists in large parts of new sets of results from an extended analysis, and does not in my view do much to aid the reader in the interpretation of the study. Throughout the paper, there is a general lack of separation between qualified speculation, knowledge based on existing studies, and conclusions substantiated in the analysis.

There are many numerical quantities in this paper that are given importance (correlations, averages, anomalies etc). In my view, it is not always clear what these actually are - e.g., what is being averaged and over what domain, which variables are being correlated, how averages are computed, etc. It is important that such ambiguities are addressed in a final version of the manuscript (but I think this should be fairly straight forward to remedy).

I am hesitant to recommend the publication of this manuscript, and I would not recommend its publication in its current form. Yet, I believe that the authors have done some very interesting work here, and I think that this paper could be a valuable contribution to literature - but in my view, this would require a substantial effort in revising the paper.

I have included more specific comments below, but they should not be taken as an extensive list. My main recommendation to the authors would be to focus on making it much clearer to the reader what their key findings are and how they arrive at their interpretations.

Reply: Thank you for the constructive comments. With your suggestions, we will diligently address them in our manuscript revisions, primarily by 1) distinguishing between the results and discussion sections, and meticulously organizing the main findings. Any portion of the discussion section that bears resemblance to the analysis of results will be relocated to the results section, while speculations based on data and

literature in the results section will be appropriately transposed to the discussion section. 2) Elaborating on numerical quantities, such as correlations, averages and anomalies, etc., with detailed and clear descriptions. 3) Checking and revising any ambiguous or imprecise descriptions in the manuscript, and restructuring lengthy sentences that commence with related words, to ensure clarity and precision. In conclusion, we will improve the text so that the reader can clearly understand the main findings and interpretations of this manuscript.

**## MAJOR COMMENTS**

There are many statements through the manuscript of 2020 being a year of very high sea ice exports through the gateways. The clearest example is perhaps L443, which states that area flux through the gateways was "extremely large compared to the 1988-2020 climatology". I find that this is an inaccurate description of the data as described. E.g. in figure 3: 2020 looks like a year of fairly high exports, but does not seem to particularly stand out; for example, area exports were higher in all three gates preceding year of 2019. Elsewhere, it is stated that the JFM 2020 area flux through the FS gateway was 1.2 times the 1988-2020 mean (L219) - this does not strike me as extreme given the large interannual variability (and there is a corresponding ~20% *negative* SIAF anomaly through the FJL-NZ gate in 2020). Similarly for "extremely low" chlorophyll concentrations in April (L20) - is this really substantiated by the analysis? I think it is important to be quite careful with language here; strong statements need to be supported by a clear justification based on underlying data. (Note: I do not think this season being less "extreme" means that it is not worth studying! For example, an observation that an extreme AO year only had a moderate impact on sea ice outflow would in my view be a valuable contribution.).

Reply: Thanks for the suggestion. According to Figure 3, the positive SIAF anomaly for 2020 is large but indeed not particularly prominent, so we will check the statement that 2020 is a very high year for sea ice export and revise this description to make it accurate. For the description of the "extremely low" chlorophyll concentrations, we will calculate a time series of spatially averaged chlorophyll concentration over the BGS for 2005-2020 to give an accurate description based on the data.

The Results and Discussion sections in my view need reorganisation. There are many instances where some fairly strong claims which are not based on the present analysis are included in Results. Conversely, much of the Discussion section consists of the presentation of whole new sets of results and analysis that have not previously appeared in the manuscript. I strongly suggest reorganising the manuscript such that the Results section is reserved for the presentation of the outcome of the analysis (including that of

SST and chlorophyll), and the Discussion section for the authors' interpretation of the data/comparison with literature/qualified speculation etc. Given the complex topic and the many different datasets involved, the Discussion section of this paper is a great opportunity to carefully guide the reader through each key argument with reference to results from the different analyses.

An example of discussion in the Results section is found in ~L288 - L293. Here, there are some fairly broad statements about stratification which are not actually substantiated in the data. Similarly with e.g., L233-L235 / L239-241 / L299-L301 etc. These are important points but they require explanation and are not in my view obvious from the data alone.

An example of results in the Discussion section is the correlation analysis in 4.1. This reads very much like new results to me. Same with the SST/chlorophyll analysis that follows.

Reply: Thanks for the constructive suggestions. We examined the results and discussion sections of the manuscript and, as you say, they appear to have blurred boundaries and need to be reorganized so that the reader can easily understand. For the Results and Discussion sections, we will carefully check and place them in the appropriate section. We will place the correlation analyses of 4.1, SST and chlorophyll in the Results section. We will move speculations and statements made in the Results section of 3.1-3.4 based on data or literature (L288-L293/L233-L235/L239-241/L299-L301, etc.) to the Discussion section, and provide detailed explanatory notes for these statements.

If I have understood correctly, the atmospheric/sea ice area/chlorophyll analysis was done using averages across the area defined on L82 and shown as black polygons in Figure 1 BGS (if this is *not* the case, I suggest making it clearer which regions are used). The GS and BS boxes cover a vast area spanning quite different climatic environments and quite different ecosystems. I would guess that about half of this area is more or less never ice-covered in the present epoch; if chlorophyll in the southeastern part of the domain is influenced by sea ice inflows it may be in a rather indirect way. Likewise, if Figure 7 shows surface heat budget for the entire BGS area, I am not sure how meaningful they are. The authors need to justify this choice of a study area and discuss the implications of an analysis spanning widely different domains (or, if I have misunderstood, I would suggest that they clarify their methods..).

Reply: Yes, we used the averages within the black polygons in Figure 1 for the atmospheric/sea ice area/chlorophyll and surface heat budgets analysis. We have identified an irrational division of the study area and will change the boundary between the east and west sides of the study area to the coastline in the revised manuscript. The implications of this analysis will be discussed in the context of the fact that there will be ice-free areas within the study area. For chlorophyll, in conjunction with the

recommendations that follow, we will remove this part of the discussion. For surface heat budgets, we will recalculate surface heat flux anomalies and their associated changes in sea ice thickness, not across the entire BGS area, but in the ice-covered areas within the BGS area.

I would recommend going through the manuscript in general for clarity. In quite a few instances, the meaning of a sentence is ambiguous or, I suspect, not in line with the intended meaning. I've included some examples below under "technical/language", but they do not constitute a complete list.

Reply: We will revisit the manuscript to avoid ambiguities and incorrect meanings. Thanks for the reminder.

I found the discussion of ecological conditions/phytoplankton to be severely lacking. First, it looks like the discussion is based on satellite chl-a concentrations from areas not directly influenced by sea ice (spring blooms in southern BS, eastern/southern GS). It was unclear to me whether the authors believe that conditions were favourable (L398) or unfavourable (L19) to biological production. Repeated statements about extremely low chlorophyll-a in April 2020 seem to refer to sea ice-covered areas where there are in fact no satellite measurements (Fig 9a). The statement that "phytoplankton seeding" and "residue of marine nutrients" were responsible for high primary production (L399-L400) is neither explained nor substantiated at all. I would suggest removing the chlorophyll discussion from the manuscript unless it is completely overhauled.

Reply: Based on chlorophyll concentrations obtained from satellite remote sensing products, we quantified and analyzed the relationship between chlorophyll and sea ice area over the BGS. Due to the limitations of the data, we attempted but could only do some simple analyses and we were unable to substantiate and explain further mechanisms and linkages, resulting in a serious lack of discussion on ecological conditions/phytoplankton. We will therefore accept your suggestion to remove the discussion of chlorophyll from the manuscript.

I would say something along the same lines about the section on surface heat fluxes (L304 - L323). First, there is the issue of (apparently) using integrated fluxes over areas including huge ice-free areas of the north Atlantic to assess ice melting in the small northern/westernmost portions the domain (I would expect there to be large heat losses from the ocean to the atmosphere in the southern BS and over the West Spristbergen Current in winter, for example - how do they affect this estimate?). Second, the values that are given for estimated ice thickness change spans a huge range (1 to 41 cm). Lastly, I think it would be quite helpful to plot the actual delta h alongside the actual fluxes to clarify what we are actually looking at.

Reply: 1) We realize that it is not reasonable to assess the effect on sea ice melting by the surface heat flux anomalies across the entire BGS area, as this incorporates heat losses from the ocean to the atmosphere. We will change the calculation area for surface heat flux anomalies to the ice-covered area in the corresponding year to assess the impact of surface heat flux anomalies on sea ice melting. 2) We will then recalculate the estimated reduced ice thickness and superpose it on the surface heat flux anomaly graph, i.e., Figure 7.

**## MINOR COMMENTS**

It might be nice to show time series of AO/CAI for context since these are pretty central to the paper. It's not strictly necessary, but I would suggest adding this as a small figure, at least in the supplementary.

Reply: Following your suggestion, we will add the AO/CAI time series to the appendix.

It is stated in the abstract (L14) and elsewhere that "the variability of.. total SIAF was dominated by changes in ice motion speed", listing a high and significant correlation. In the text I could not find any details about how you actually calculated this, and given that this is a key point in the abstract I think the authors need to elaborate on actually how this quantity is computed, e.g.:
Is this a correlation between monthly values of a) SIAF across all three gates and b) mean speed across all three gates? Over what time period?
How are no-ice instances counted (for SIM and SIAF)? As zero, or are they not included?

Reply: We will add a description of the calculation of this correlation in the method section. 1) Actually, this is about the correlation between the sum of monthly SIAF for all three gates and the monthly mean SIM speed for all three gates. Seasonal mean SIAF and SIM speed were calculated based on winter (JFM) and spring (AMJ) to further quantify the correlation between them seasonally. Correlations are calculated based on data from 1988 to 2020. 2) NSIDC SIM speed is only available when the SIC is greater than 15%, and SIAF will be negligible when the SIC is zero.

Along similar lines, it should be clearly stated whether "sea ice thickness anomalies" include "thickness" from ice-free periods. (Fine either way, but "thinner ice" and "less frequently ice-covered waters" are different things, for example).

Reply: Thanks for the suggestion. We will add sentences to the section on sea ice thickness anomalies to clarify whether such anomalies include "thickness" from ice-free periods. In the calculations, "sea ice thickness anomalies" do not include the "thickness" from ice-free periods.

I think it should be discussed whether comparing trajectories from one single day (e.g. 31 May 2020) with a climatological mean vector field (e.g. May 1988-2020 mean drift) may (or may not) influence the analysis. Is a trajectory along an average field different from a time average of several trajectories? Does the long-time average introduce a low speed/short trajectory bias? (I don't really know - but I think it warrants at least a brief comment in the paper given that this is an important point in your study).

Reply: This is an important point. Yes, this may be a misunderstanding due to the lack of clarity in the labeling of Figure 4, which will be revised to clarify its meaning. The trajectories in Figure 4 all refer to the backward trajectory from a particular day. Figures 4a-c all refer to the backward trajectory from a specific date in 2020 back to January 1. The 1988–2020 mean trajectories in Figures 4d-f do not refer to the temporal averaging of multiple trajectories, but rather the average SIM vector field in 1988–2020.

SIAF is basically the integrated product of SIM and SIC, so if it were not SIM that controlled SIAF variability it would presumably be SIC (or am I misunderstanding?). You should therefore probably include the corresponding correlation between SIAF and SICs somewhere in the paragraph at L236 in support of your statement that SIM controls SIAF.

Reply: We will calculate the correlation between SIAF and SIC and state their correlation in the corresponding text.

L45: I would revise the statement that sea ice outflow "contributes" to deep water formation in the north Atlantic. Please clarify what is meant here (presumably that high sea ice export -> high fw input -> increased stratification -> *inhibited* dw formation - or am I missing something?). I also can't see that Lemke et al. 2000 is a great reference here - they show a large variability in sea ice export but don't actually look at DW formation as far as I can tell?

Reply: Good suggestion. We will revise this sentence to clarify the effect of sea ice outflow on the deep water formation in the North Atlantic. And we will remove this literature and cite more relevant references to support our statement.

Figure 1 is excellent. The authors could consider adding mean SIC or similar (not necessary but might give some additional context to readers unfamiliar with the region).

Reply: Considering readers unfamiliar with the region, we will add the average SIC for January-March 2020 as a background to Figure 1.

The switch from CS2SMOS to PIOMAS seems to warrant a (quick) comparison of the two in the overlapping period. Should add at least a sentence about this somewhere around L123.

Reply: We will compare these two SIT products in the overlapping period and explain this in the data section of SIT.

Which temperature does the oiSST give for fully ice-covered waters? What about for partial ice concentrations? Should be included in the presentation of this product somewhere after L124, and may be relevant for the interpretation of the SST data.
Reply: Thanks for the suggestion, we will add the corresponding details in the data section of the SST.

L159: I believe that there is no SIM vector for SIC<15% in this product? I don't believe that is a significant problem for your methods, but it is probably worth a mention here.
Reply: We will add specific notes in the methods section.

L161: Was one trajectory computed from each grid point on each gateway?
Reply: In fact, we divide a total of 400 points from the location of the three passageways and then calculate the backward trajectory from these points. The number of points varies for each gate, from 200 points (with a distance of ~ 448 km) for the FS, 100 points (with a distance of ~ 284 km) for S-FJL and 100 points (with a distance of ~ 326 km) for FJL-NZ.

L169: This seems to imply forward propagation, I would suggest reordering the equations to show backward propagation if that is what you do. Also, if delta t is negative as suggested in the preceding paragraph, I think this equation is in fact incorrect (wrong sign of second term)?
Reply: Thank you for the reminder. We will check and revise the formulas to show backward propagation.

L223. If you attribute the 20% positive anomaly at FS to the AO anomaly, it seems strange to say that 20% negative anomaly at FJL-NZ means that flow across this gate was "not sensitive". Could it not be a similar-magnitude response of opposite sign? Please elaborate.
L226: See above. This seems like a huge negative anomaly, and the idea that positive=response and negative=insensitivity needs an explanation at least. I also have a hard time seeing the 85% negative anomaly in Figure 3c - is this because the absolute values are small?
Reply: 1) As mentioned in the result section, positive AO anomaly promotes sea ice transport to the BGS, so we assume that a response to positive AO anomaly implies an increased SIAF, which would be large compared to the 1988-2020 climatology. In contrast, the FJL-NZ gate showed a large negative SIAF anomaly, which we believe

that the SIAF through the FJL-NZ does not respond significantly to the positive AO anomaly. We will revise the statement about the response being sensitive or insensitive. 2) L226 refers to the fact that the spring cumulative SIAF through the FJL-NZ gate is only 14.1% of the 1988-2020 climatology, while Fig. 3c is the difference between the monthly SIAF and the 1988-2020 climatology. so it is difficult to see the 85% negative anomaly there.

L230: The phrasing here is a bit ambiguous - should make it clear what exactly these percentages are. (I assume that they are the fraction of SIAF through all three gates that went through FS - but it can read as the percentage of the *anomaly*). Same for L13.
Reply: We will revise the phrasing to clarify the meaning, these are the SIAF through a single gate as a fraction of the SIAF through all three gates.

L233: In my view, this is a strong statement that needs substantiation. What is your evidence that the low Arctic SIA was a result of increased outflow? Probably also a better fit in Discussion.
Reply: We will quantify the correlation between Arctic SIA and SIAF and discuss it in an updated discussion section.

L237: Please explain exactly what this correlation is.
Reply: A detailed explanation of this SIAF and SIM correlation calculation will be added in the method section.

L251: " the anomalies of sea ice volume outflow.." This is a bit confusing. You don't have these numbers, right - is it that you would *expect* the volume export anomaly to be more pronounced than the area export anomaly? Rephrase for clarity.
Reply: We will rewrite this paragraph for clarity.

L267-L268: This seems true for the FS, but not for the two other gates?
Reply: We will revise the sentence to state that it only applies to the Fram Strait.

Figure 4: I suggest using much thinner lines to show that these are trajectories and not a continuous scalar field - I think this figure is nice, but it took me quite a while to figure out what was going on.. Given that trajectories don't run past January, I don't think the colour by date actually adds that much, so I wouldn't be worried if thinner lines show the colours less well. Should also change the labels in abc from e.g. "April 2020" to "30 April 2020" if you indeed only show back-trajectories from one single day in these. (You should also explain why you chose to compare one day per month with monthly climatological means).

Reply: Thank you for your suggestion. 1) We will use thinner lines to show the trajectories, these dense lines are because we used 400 points located at three gates as endpoints. 2) The trajectories in Figure 4 all refer to the backward trajectory from a particular day. Therefore, we will change the labels of Figure 4 to specific dates for easier understanding. Specifically, figures 4a-c all refer to a backward trajectory from a particular date in 2020 back to January 1. The average 1988-2020 trajectory in Figures 4d-f does not refer to the temporal averaging of multiple trajectories, but rather the average SIM vector field in 1988-2020.

L298: It's not clear to me from Figure 6a that there was a positive SIT anomaly in the BS - please explain.
Reply: The positive SIT anomaly in BS in Figure 6a is small and not very obvious, and we will revise the description of this sentence.

L301: Similarly, it does *not* look to me like an overall positive anomaly in the GS in Figure 6d. And it seems strange not to address the east-west pattern in the GS here. And, as mentioned elsewhere, I think you also need to say something about whether the positive anomalies are a result of anomalously high sea ice *extent* or anomalously thick ice since this could impact your argument about increased import of thicker ice from the central Arctic (probably depends on how you compute these means).
Reply: We will revise the phrasing of this sentence and discuss the east-west pattern of the SIT anomalies that occur in GS. We will also clarify how SIT anomalies are calculated and explain the meaning of positive anomaly results.

L337-8: Is it surprising that you did *not* find a significant correlation between SIAF and AO in months other than February? Seems to warrant at least a mention.
Reply: We will add sentences stating that no significant correlation was found between SIAF and AO except for February.

L340: What about R?
Reply: The correlation coefficient R is shown in Table 2 and we will add the description of R to this sentence.

L345-351. I found it difficult to see this in Figure A1ab, please explain how you arrive at this conclusion. To me, it looks like there is little difference except perhaps that the drift is stronger in the negative AO phase (A1b). It also looks like these are the trajectories of ice arriving at the gate in mid-summer - so is it actually representative of AO anomalies in winter? Please clarify.
Reply: 1) This can be seen at the end of the backward trajectory in Fig. A1 ab (blue

trajectory). The western edge of the blue trajectory in Fig. A1a extends westwards, while that in Fig. A1b moves closer to the prime meridian. This suggests that under the positive phase of AO, sea ice originates further west, i.e., the TPD shifts westwards, thus reducing the spatial extent of the BG. Rigor et al., (2002) mentioned that such a pattern favours the transport of sea ice from the central Arctic Ocean to the BGS. The results in Fig. A1 ab are consistent with the westward alignment of the TPD under the positive AO in Figure 4. 2) These represent the backward trajectory of ice arriving in the Fram Strait on 30 June back to 1 January, and we focus on the location of the winter (JFM) trajectory which is also the source location of the sea ice arriving in the Fram Strait. We have chosen years with winter AO anomalies to check whether the January-June backward trajectory of sea ice arriving in Fram Strait in June is affected by winter AO anomalies. We will revise this paragraph to clarify.

Reference:

Rigor, I.G., Wallace, J.M., and Colony, R.L.: Response of sea ice to the Arctic Oscillation, J. Clim., 15(18), 2648–2663, https://doi.org/10.1029/1999gl002389, 2002.

L365: Are these SST anomalies integrated over the whole area? Also, are the maps in Figure 8 really anomalies from the climatological mean for the month? To me they look like anomalies from the SST average across the entire season or something (+4C anomalies in the southern BS in July and -3C anomalies in the northernmost range in April both strike me as weird). I could certainly be wrong - but it might be a good idea to double check.

Reply: Thanks for the reminder. These SST anomalies were obtained by performing calculations within the study area, and we will check the process of calculating SST anomalies and redrawing Figure 8.

L389 onward: Again, it seems strange to me to do this sort of analysis across this very large area - you at least need to discuss the implications of 1) including large perennially ice-free areas, and 2) there being no satellite measurements of chlorophyll in ice.

Reply: Thank you for your suggestion. In light of the major comments related above, we acknowledge that the discussion of chlorophyll is indeed severely lacking. Given that only satellite remote sensing chlorophyll concentration products were used, we will follow your suggestion and remove the chlorophyll discussion section from the manuscript.

I found the Conclusions section to be quite clarifying and well-written. I would encourage the authors to use a similar style in an updated Discussion section.

Reply: We will use a similar style in the revised discussion for clarity.

**TECHNICAL/LANGUAGE**

L13: "77.6%": I gather from the conclusion that this is the fraction of the total flux through the three gateways that goes through the FS gate. That was not clear to me when reading the abstract alone.

Reply: We will revise the sentence to clarify that 77.6% is the proportion of the total flux through the three gates that is through the FS gate.

L51: "More primary productivities" - please use more precise language.

L55: I would suggest reordering this sentence for clarity.

L63: "Relatively extreme" - please use more precise language.

Reply: We will revise these phrases using more precise language and, for sentences, we will reorganize them for clarity.

L69: "Thereby.. ..the BGS." - Meaning of this sentence is unclear.

Reply: This sentence will be rewritten to clarify the meaning.

L82: Studies -> Study?

L107: "abnormal" - "anomaly"?

L124: "Could be used as the best proxies for.." - please rephrase. "We use SST and chl as proxies for.."?

L137: Suggest replacing "a significantly more advanced" with "an advanced" or similar ("more" relative to what?).

L143: "According to" - "In accordance with" or similar?

L146: These directions aren't really S/N? Maybe replace "southward" with "toward the BGS" or something.

Reply: These grammatical mistakes and inappropriate expressions will be revised as suggestions.

L185: "with the values maintaining the top three.." - This phrasing is a bit unclear (here and elsewhere in the manuscript).

Reply: We will revise all similar phrasing in the manuscript to clarify its meaning.

L215-L217: Suggest moving this sentence to data/methods (and adding e.g. "relative to the 1988-2020 climatology" after "June 2020" in L218).

Reply: We will follow your suggestion to move the sentence and add "relative to the 1988-2020 climatology" in the original position.

L238: Maybe "monthly SIM speed" to avoid confusion.

L256: "very analogous to" - "similar to"?
Reply: We will revise the phrasing to avoid confusion.

L261: "confluence of the Kara Sea and the CAO" - please explain.
Reply: Thanks for the reminder. We found that the geographical location of the confluence is not visible in Figure 4. We will add the location of the marginal sea and the central Arctic Ocean to Figure 4.

L262: "exhibited a relatively high tortuous feature" - unclear-please explain.
Reply: This sentence is used to describe that the backward trajectories obtained from the S-FJL gate are more curved than that obtained from the FS gates, and we will revise this phrase to make it clearer.

L280: Sentence needs fixing; meaning of "since then" is unclear.
Reply: We will rewrite this sentence for clarity.

L280-L288: I strongly suggest revisiting this entire section for clarity; I found it quite difficult to follow most of it.
Reply: For this section, we will reorganize and rewrite it to ensure that this section is clear and easy to understand.

L287: What is this correlation? Is it between the sum of the SIAF through all three passageways and SIA in the BGS? Also: "has been identified" should be rephrased for clarity (e.g. "we found a correlation.." if that is the meaning).
Reply: This refers to the correlation between the sum of the SIAF through the three passageways and the SIA of the BGS. We will rephrase it to clarify the meaning based on your suggestion.

L304: "dominated" - "governed"?
L337: "most active" - rephrase for clarity.
L347: "spatial scope" - "spatial extent"?
L376: typo (southern/northern).
L396: "a single SST" - rephrase for clarity.
Reply: We will correct misspellings and inappropriate phrases based on your suggestions and rephrase inappropriate expressions for clarity.

I suggest looking over sentences starting with "thereby", "furthermore" etc. and checking that these preserve the intended meaning.
Reply: We will check and revise the entire manuscript to see if the meaning of these

sentences retains the intended meaning.

Figure A1: Please label clearly which plots correspond to positive/negative phases, or at least explain/refer to labels in the caption ("AO+sd" is not very intuitive on its own). I would also suggest adding the number of years that go into the mean (e.g. "n=3") somewhere on each subpanel (not necessary but would aid interpretation).

Reply: Thanks for the suggestion, we will explain the meaning of the label (e.g. "AO+sd") in the caption of Figure A1 and add the number of years into the mean calculation in the sub-panels of Figure A1.

---

## Author Comment (AC4)

**Response to RC4**

We express our sincere appreciation for the time and valuable feedback provided by the reviewer on the manuscript titled "Impacts of anomalies in Arctic sea ice outflow on sea ice in the Barents and Greenland Seas during the winter-to-summer seasons of 2020". All comments will be carefully reviewed, and we are committed to incorporating the suggested changes to ensure that our manuscript is significantly enhanced.

**General comments:**

1. What is the reason to choose these two atmospheric circulation patterns for assessing the effects of large-scale atmospheric circulation on the changes in Arctic sea ice outflow? I would suggest that the authors also look at whether there is an abnormal North Atlantic Oscillation in 2020, and if so, the influence of the North Atlantic Oscillation on Arctic sea ice outflow should be discussed.

Reply: Thanks for the suggestion. Based on previous literature (Kwok, 2009; Vihma et al., 2012), the main atmospheric circulation patterns that have an impact on Arctic sea ice outflow are Arctic Oscillation (AO), North Atlantic Oscillation (NAO), and Central Arctic west-east air pressure gradient Index (CAI). we chose AO and CAI mainly because we found that they showed obvious positive anomalies in the winter (JFM) and spring (AMJ) of 2020, with positive values ranking high in the period 1979-2020 (Table 1), while NAO did not show such strong anomalies in 2020. Certainly, the impact of NAO on the Barents and Greenland Seas (BGS) cannot be ignored, and we will quantify the correlation between NAO and sea ice conditions in the BGS and add a corresponding discussion.

References:

Kwok, R.: Outflow of Arctic ocean sea ice into the Greenland and Barents Seas: 1979–2007, J. Clim., 22(9), 2438–2457, https://doi.org/10.1175/2008jcli2819.1, 2009.

Vihma, T., Tisler, P., and Uotila, P.: Atmospheric forcing on the drift of Arctic sea ice in 1989–2009, Geophys. Res. Lett., 39(2), https://doi.org/10.1029/2011gl050118, 2012.

2. Section 3.3 focus on the comparison of the reconstructed sea ice backward trajectories in 2020 with the 1988-2020 climatology. The comparison assumes that the reconstructed sea ice backward trajectories are convincing. However the validation of the reconstructed backward trajectory method is not sufficient. I would suggest that the authors provide more assessments on the validity of the reconstructed trajectories using buoy observations.

Reply: To give credibility to our sea ice backward trajectory reconstruction results, we plan to use observed buoy data for validation, such as MOSAiC buoy data and buoy observations from IABP (International Arctic Buoy Programme). We will evaluate the

effectiveness of the reconstructed sea ice backward trajectory considering the case where AO and CAI are in the positive or negative phase.

3. Section 3.4 discussed the anomalies of the sea ice area and thickness in the Barents and Greenland Seas. The data analysis on sea ice area is relatively adequate. However the analysis of sea ice thickness anomalies is mostly qualitative. I would suggest that the authors provide more quantitative results on sea ice thickness anomalies and discuss them in details.

Reply: Thank you for the suggestion. We will provide a quantitative analysis of sea ice thickness anomalies in the BGS and further discuss them based on the data results.

4. If the abnormal Arctic Oscillation and Arctic sea ice outflow do not occur in winter, but in other seasons, will the effect be different for the ice and marine environment conditions in the Barents and Greenland Seas? It would be better to add more discussion on this.

Reply: Following your suggestion, we will add a discussion on the impact of summer AO anomalies on the BGS, since generally, sea ice motion responds more strongly to the atmosphere in summer. We plan to quantify sea ice area flux (SIAF) through Fram Strait and sea ice conditions in the BGS using years with positive summer AO anomalies during 2010-2020, to explore the impact of summer AO anomalies on sea ice and marine environment conditions in the BGS.

5. It is a last resort to use different sea ice thickness products (radar altimeter and PIOMAS model-based data) in different seasons. My concern is whether using different data produces inconsistent results. For example, during the freeze-up period, whether there is deviation or even contradiction between the qualitative conclusion and quantitative results using PIOMAS model-based data and radar altimeter?

Reply: This is a good point. We will calculate the results of sea ice thickness (SIT) anomalies for both SIT products in the BGS during the freezing period and compare whether there is a large discrepancy between SIT anomalies obtained from these two products during the same period. And we will add sentences to assess the consistency of the two datasets and whether there is an impact on the relevant conclusions.

**Specially comments:**
1. Line 72, change "during winter-spring 2020" to specific month, since you do not define the range of winter and spring months before that.

Reply: We will revise this sentence to clarify the specific months to which winter and spring refer.

2. Line120-122, "We regridded the monthly SIT data on the 25-km EASE-Grid to maintain consistency with the CryoSat-2/SMOS SIT data." These two datasets also have different temporal resolutions. How was this difference addressed in your study?

Reply: We obtained the same temporal resolution as the PIOMAS SIT data by monthly averaging the weekly CryoSat-2/SMOS SIT data. We will add a description of how to harmonize the temporal resolution in the SIT data section.

3. Line 161, change "restructured" to "reconstructed" to unify the expression and apply to the entire manuscript.

Reply: We will check the entire manuscript and revise this inappropriate expression.

4. Line 271, "enhanced sea ice meridional motion", remove the "meridional", since you do not directly calculate the meridional sea ice motion speed.

Reply: We will remove the "meridional" as you suggested.

5. Line 294, The text on the right side of Figure 5 is too busy and not intuitive enough. It is preferable to express the trend of sea ice area graphically.

Reply: Thanks for the suggestion, we will use charts to depict the trends of the sea ice area for a more intuitive understanding.

6. Line 319, "The anomalies in cumulative surface heat fluxes from January to June 2020 can be related to a reduced decrease of 0.01–0.41 m in SIT, estimated using the Eq. 4" This is ambiguous. Does cumulative mean cumulative over time or across net surface heat fluxes?

Reply: In fact, it refers to the accumulation of the entire net surface heat fluxes containing surface heat fluxes of sensible heat, latent heat, net longwave radiation, and net shortwave radiation. We will revise the sentence to clarify the meaning.

7. Line 327, change the y-axis title in Figure 7 to "Surface heat fluxes anomaly", because it is surface heat fluxes anomaly instead of surface heat fluxes in the figure caption.

Reply: We will revise the y-axis title for clarity.

8. Line 377-378, "In addition, we examined the statistical relationship between the April SIA and the monthly SST with a lag of 1–3 months in the BGS (Table A4)." Is this correlation for the detrended SIA and SST? This comment applies to the entire manuscript.

Reply: In our manuscript, the correlation results are all correlations between the detrended variables. We will add notes to make them clear.

9. Line 388, "The Chl-a over the southern Greenland Sea in April 2020 was smaller compared to the previous 5 years." Give the latitudinal range of the southern Greenland Sea.

Reply: Thanks for the reminder. Our revised manuscript will specify the latitude range over the southern Greenland Sea.

10. Line 389, "A significant negative correlation between Chl-a and SIA in April over 1998–2020 was identified". The geographical scope of this sentence is unclear. Is there a negative correlation between Chl-a and SIA in the BGS or only in the Greenland/Barents Sea?

Reply: The negative correlation between Chl-a and SIA is significant only in the southern Greenland Sea, and we will revise this sentence to clarify the meaning.

11. Some illustrations need to be further revised, for example, the definition of the study area in Figure 1 is not so normative.

Reply: Thanks for the suggestion. We will revise the definition of the study area in Figure 1 to make it reasonable and normalized. Accordingly, we will update the analysis results associated with it.

---

## Author Response (AR1)

**Response to Editor**

Dear Dr. Homa Kheyrollah Pour,

Thank you for handling our manuscript. According to the comments from the anonymous reviewer, we have made a thorough revision of our manuscript. Please see below for a short summary of our revision:

1) We added additional calculation and discussion according to the comments of the reviewers.

   In the Data section, we added an assessment of consistency for different sea ice thickness products; In the Results section, we proved the effectiveness of the reconstructed results of the ice backward trajectory; In the discussion section, we added the impact of abnormal AO and CAI on sea ice thermodynamics, the impact of low Arctic sea ice outflow on sea ice condition, and how summer AO affected the Arctic sea ice outflow.

2) We reorganized results and discussion in accordance with the reviewers' suggestions.

3) Based on the reviewers' comments, we removed the chlorophyll section of the Discussion and the recommendations section of the Conclusion.

4) We corrected the calculation issues raised by the reviewer and revised sentences and figures according to specific comments from the reviewers.

Please find the following files in our submission package:
1) The clean manuscript, 2) the manuscript with tracked changes, and 3) the response letter.

Below are our point-by-point responses (black text) to the comments (blue text) from reviewers. Line numbers refer to the revised manuscript with track changes.

Thank you for your time.

Best regards,
Ruibo Lei and co-authors

**Response to RC1**

Thank you for your time and constructive comments on the manuscript "Impacts of anomalies in Arctic sea ice outflow on sea ice in the Barents and Greenland Seas during the winter-to-summer seasons of 2020". We have considered each comment carefully and modified our manuscript accordingly as part of this revision.

**Major comments**

1Title:

I would like to see a better title. The current title read: "Impacts of anomalies in Arctic sea ice outflow on sea ice in the Barents and Greenland Seas during the winter-to-summer seasons of 2020" When I look the following text, I felt the manuscript is mainly dealing with the factor that create the Arctic sea ice outflow anomalies, as author stated in the abstract (L9): "the impacts and feedback mechanisms on a seasonal scale of anomalies" So, I suggest authors to speak out what "impacts", e.g. atmospheric circulations. One possibility could be: The impact of atmospheric circulations on the anomalies of sea ice outflow and their feedback mechanisms in the Barents Sea and Greenland Sea

Reply: According to the suggestion, we revised the title to "The impacts of anomalies in atmospheric circulations on Arctic sea ice outflow and sea ice conditions in the Barents Sea and Greenland Sea: case study in 2020". (Line 2-4)

2 Abstract:

There are totally 322 words. I think it is too long, please compact it to e.g., 250 words. However, if TC accepts a long abstract, so be it, but please add some compact analyses/statement to echo latest state of the art findings. I am sure there are papers dealing with Arctic sea ice outflow

Reply: We added an analysis of the effect of atmospheric circulation patterns on Arctic sea ice outflow to echo the study by Sumata et al. (2022). (Line 17-21)

3 Introduction:

This part is largely ok, but as I have stated in previous point, please consider echoing your work with UpToDate finding. I recommend authors to check Sumata et al., 2022. This paper should be cited in your work. Some comparison would be even better in results/discussion section. The language can still be improved. This comment valid for entire manuscript.

Reply: In section 4.3, we added a comparison of the changes in sea ice and oceanic conditions over the BGS compared to the climatology in a scenario with relatively low

Arctic sea ice outflow (2018) as discussed by Sumata et al. (2022). We improved the expression and language throughout the manuscript. (Line 518-527)

4. Data and method:

I am quite impressed that such comprehensive data sets are used in this work, well done. I wish authors could make further elaborate on data accuracy and comment/assess the data consistency, for example, authors wrote "Here, we used the CryoSat- 122 2/SMOS SIT from December to April, and the PIOMAS SIT from May to June in 2011–2020 to estimate the anomaly in SIT during the study year of 2020". Do I need to worry about the inconsistency of the data sets applied here?

Reply: We added an assessment of data consistency for CryoSat-2/SMOS and PIOMAS SIT products to the Data section (Line 141-149). The results showed that the spatially averaged differences between these two SIT anomalies from December to April are about 6.0%-13.3% of the monthly magnitude. The correlation between SIT anomalies calculated from them is 0.95 in 2011–2020 ($P<0.05$). We thus believed that the differences are acceptable for calculating SIT anomalies.

Line: 180-182: "We note,,,". So, this do suggest that in study deals with the impact of the atmosphere on sea ice not ocean at all. I suggest authors express this argument explicitly already in the beginning, e.g., introduction.

Reply: We expressed the argument in the introduction. (Line 84-87)

5 Results:

Figure 2 is very comprehensive and informative, yet in the main text, I see only once Figure2 (the first column of Figure 2), please add more instruction on what text explain/analyses other columns of the Figure 2.

Figure 3 is also very informative; I suggest you separate last row of each panel to make it clearer and easier to distinguish from others. Furthermore, any patterns can be extracted from this figure?

3.3 section is very interesting. However, in order to prove the effectiveness of the reconstructed results of the ice floe backward drift trajectory, it would be interesting to compare the ice floe backward drift trajectory with forwarded observed buoy drift trajectories for example, under the scenarios of AO+, AO-, CAI+, CAI – and see whether the buoys' drift trajectories are consistent with the reconstructed results. If not, any impact on your results and conclusions

Could you elaborate further whether or not the abnormal AO and CAI would have impacts on sea ice thermodynamics, e.g. total ice mass balance before ice floes reached the Fram Strait? I would like to see more discussions.

Reply: We revised the manuscript and the following is the corresponding response:

1) We added descriptions derived from Figure 2. (Line 234, 239-240, 257, 441, and 466)
2) We separated the last row of each panel of Figure 3 and added a short text to highlight the knowledge obtained from Figure 3. (Figure 3, Line 279-285)
3) Using the buoy data from MOSAiC and International Arctic Buoy Program, we verified the validity of the reconstruction results of ice backward drift trajectory for the study year 2020, as well as for years with AO+, AO–, CAI+, and CAI–. In these cases, the distance and direction of the reconstructed backward trajectories are consistent with the buoy trajectories. (Line 309-322)
4) We added a discussion of the influences of abnormal AO and CAI on sea ice thermodynamic process in section 4.1. Only in the AO+ case, the Freezing Degree Days (FDD) along the backward trajectories was lower than the 1988–2020 average, which is not favorable for sea ice growth. In the AO+ and CAI– cases, ice floes stayed more time south of 82°N before reaching Fram Strait than in the other cases. (Line 424-435)

6 Discussion:

Please strengthen the linkages of your work with the latest state of the art of research e.g., Sumata et al., (2022). In such extreme season, what are the possible impacts of Arctic sea ice outflow on the a) sea ice state and b) marine hydrographical and c) ecological conditions in the Barents Sea and Greenland Sea?

Reply: We added a comparative discussion in section 4.3, using 2018 as mentioned in Sumata et al. (2022), to obtain the impact of abnormally low Arctic sea ice outflow on sea ice and oceanic conditions in the Barents and Greenland Seas. (Line 518-527)

7 Conclusions and recommendations

I suggest you drop recommendations because you merely "recommended to further collect the in situ observation,,,, in the study region" which is not necessarily entitled as recommendations, unless if you recommend some specific concrete parameters/variables or some specific instrumentation to be observed or to be used and further to be linked to each other.

Reply: We removed the original recommendations and added a more detailed recommendation as suggested. (Line 584-590)

**Minor comments**

2) Figure 1: There is no need to define a geometrically regular study regions for the BGS. Its northern boundary can be consistent with the defined passageways, and the area bordering Greenland and other islands can be consistent with the shoreline.

Reply: We redrew the Figure 1 and changed the defined study area of the Barents and Greenland Seas as suggested (Figure 1, Line 104). The corresponding results have been updated.

3) Line 107 "In addition, we used data from the NSIDC Sea Ice Index version 3 (Fetterer et al., 2017) to obtain monthly SIA changes in the Northern Hemisphere in 2020." The purpose of using this data is unclear. In addition, data from the Arctic should be used instead of data from the Northern Hemisphere.

Reply: Considering that the correlation between the Arctic SIA and the BGS SIA was not significant, we removed the part of Arctic SIA.

14) Section 4.3, 1) also consider the scenario with low Arctic sea ice outflow, 2) Does North Atlantic Oscillation have a significant regulatory effect on the marine environment of BGS?

Reply: 1) We added a discussion of the scenario with the low Arctic sea ice outflow in section 4.3. (Line 518-527) 2) We added a simple discussion of North Atlantic Oscillation in section 4.2, mainly quantifying the relationship between the NAO index and the sea ice area in the BGS. (Line 460-468)

1) Line 50: "plays a crucial role in shaping the icescape in this region"-- change the "shaping" to "proving the preconditions"

Reply: Changed as suggested. (Line 56)

4) Line 205 "regulating the sea ice outflow from the TPD region to the BGS" change to "regulating the sea ice outflow from the Arctic Ocean to the BGS"

Reply: Revised as suggested. (Line 245)

5) Line 234 "resulting in relatively low SIAs of" change to related to..., Arctic sea ice outflow is only one of the factors affecting the reduction of Arctic sea ice.

Reply: We removed the part of Arctic SIA and deleted this sentence.

6) Line 266 "was insensitive to the changes in the TPD intensity or the CAI pattern" delete "the TPD intensity or" because you did not directly quantify the strength of TPD.

Reply: Revised as suggested. (Line 338)

7) Line 309 "the monthly surface heat fluxes" (and also other text) change to "the monthly atmospheric surface heat fluxes"

Reply: Changed as suggested in the entire manuscript. (Line 162, 214, 218, 470, 506, and 569)

8) Line 323 "sea ice during winter and early summer 2020." change to "sea ice during spring and early summer 2020."

Reply: We moved this part to section 4.2 and revised it as suggested. (Line 459)

9) Line 370 "the absorption of incoming solar radiation" delete the "incoming".

Reply: We moved this part to section 3.4 and revised it as suggested. (Line 385)

10) Line 376 "are larger in the southern BGS (76°–80°N) than in the northern part (72°–76°N)"-- this should be a mistake.

Reply: Revised it as suggested. (Line 390-391)

11) Line 395 "the complex interactions between SST, SIC and Chl-a" change to "the complex interactions between SST and SIC".

Reply: Considering the lack of discussion of chlorophyll and the suggestions of RC3, we deleted this section in the revised manuscript.

12) Line 401 "the year" change to "the study year".

Reply: Changed as suggestion, but the section containing this sentence has been deleted based on suggestions in the revised manuscript.

13) Line 403 "the abnormal Arctic sea ice flow" change to "the abnormal Arctic sea ice outflow".

Reply: Same change as in the above line.

15)Tables in Appendix: consider using simple expressions to indicate different significant levels, e.g., text in bold, or Italic.

It would be nice to apply professional language service for the entire manuscript

Reply: We corrected these grammatical mistakes and inappropriate expressions according to the comments. The tables have been revised as suggested. (Line 300, 804, 807, 810, and 813)

**Response to RC2**

Thank you for your time and constructive comments on the manuscript "Impacts of anomalies in Arctic sea ice outflow on sea ice in the Barents and Greenland Seas during the winter-to-summer seasons of 2020". We considered each comment carefully, and made corresponding changes based on the suggestion.

**Minor comments:**

Lines 19–20: I think "an extremely low Chlorophyll-a concentration observed over the BGS in April" is not supported by Figure 9. The anomaly of Chlorophyll-a concentration over the BGS in April 2020 is small shown by Figure 9a.

Reply: The sentence was rewritten as "…observed over the southern Greenland Sea in April". Considering the lack of chlorophyll discussion and RC3' suggestion, we removed the chlorophyll section, including the original Figure 9.

Line 47: Change "in the BGS" to "in the Barents Sea".

Reply: We revised it as suggested. (Line 53)

Figure 2: Add ice drift velocity anomalies to the second and fourth columns.

Reply: We added the calculated ice drift velocity anomalies in Figure 2. (Line 251)

Lines 251–253: "the anomalies of sea ice volume outflow in winter–spring 2020 were expected more obvious than the SIAF anomalies" may be not true. Because the ice thickness has larger negative trend than ice concentration during winter–spring over these regions, so it may lead to the volume outflow anomaly ranking is less obvious than the SIAF anomalies.

Reply: Referring to the suggestion and relevant literature, this sentence was rewritten as "Although the source region of the ice was generally dominated by relatively thick ice, the SIAF anomalies in winter–spring 2020 was expected more obvious than the sea ice volume outflow anomalies as the negative trend of SIT in the Fram Strait (Sumata et al., 2023) is generally larger than that of SIC (Zamani et al., 2019)." Considering the low relevance to the content of the manuscript, we decided to remove this sentence from the revised manuscript.

Sumata, H., de Steur, L., Divine, D.V., Granskog, M. A., Gerland, S.: Regime shift in Arctic Ocean sea ice thickness, Nature, 615, 443–449, https://doi.org/10.1038/s41586-022-05686-x, 2023.

Zamani, B., Krumpen, T., Smedsrud, L.H., and Gerdes R.: Fram Strait sea ice export affected by

thinning: comparing high-resolution simulations and observations, Clim. Dyn., 53, 3257–3270, https://doi.org/10.1007/s00382-019-04699-z, 2019.

Lines 319–320: How to estimate SIT changes using the Eq.4 is not very clear. I think the surface heat flux anomalies shown in Figure 7 are the results for the whole BGS. Considering parts of BGS are ice-free, it is not a good idea to estimate SIT using the anomalies over the whole BGS. Using the surface heat flux anomalies only over the climatological ice-covered regions is more reasonable.

Reply: As suggested, we re-estimated the change in SIT, using the surface heat flux anomaly over the climatological ice-covered BGS (regions with the SIC above 85% for 1979–2020 climatology). In the revised version, we moved this part to section 4.2. (Line 457-459)

Line 343-345: How many years with extreme high or low are examined here?

Reply: The number of extreme high (low) years was given in appendix (Figure A2, Line 799). There are 2 (5) years of extreme positive (negative) for AO and 6 (4) years of extreme positive (negative) for CAI in the period 1988–2020.

Line 388-389: It's good to add an appendix figure to support the conclusion of "The Chl-a over the southern Greenland Sea in April 2020 was smaller compared to the previous 5 years".

Reply: The figure below shows the changes in Chl-a over the Greenland and Barents Sea in April 2005–2020. Since we finally removed the chlorophyll section in the revised version, this figure is only shown in the response.

[Figure]

Figure 1. Mean chlorophyll concentration over the Barents and Greenland Seas in April for 2005–2020. The red dashed line indicates the chlorophyll concentration over the southern Greenland Sea in April 2020.

**Response to RC3**

We truly appreciate the reviewer for the careful reading and constructive comments on the manuscript "Impacts of anomalies in Arctic sea ice outflow on sea ice in the Barents and Greenland Seas during the winter-to-summer seasons of 2020". All comments have been carefully considered and revised in the revised manuscript.

The following are our revised responses to these comments:

Unfortunately, the manuscript as a whole lacks for clarity in my view. I have read this manuscript several times, and I am still not sure I can summarise the main findings. There is a lot of interesting information in the paper, but I find it hard to follow the *reasoning* - it was often not clear to me what one should make of the specific results that were presented, and I often had a hard time understanding on what basis the authors arrived at their conclusions. The discussion section currently consists in large parts of new sets of results from an extended analysis, and does not in my view do much to aid the reader in the interpretation of the study. Throughout the paper, there is a general lack of separation between qualified speculation, knowledge based on existing studies, and conclusions substantiated in the analysis.

There are many numerical quantities in this paper that are given importance (correlations, averages, anomalies etc). In my view, it is not always clear what these actually are - e.g., what is being averaged and over what domain, which variables are being correlated, how averages are computed, etc. It is important that such ambiguities are addressed in a final version of the manuscript (but I think this should be fairly straight forward to remedy).

I am hesitant to recommend the publication of this manuscript, and I would not recommend its publication in its current form. Yet, I believe that the authors have done some very interesting work here, and I think that this paper could be a valuable contribution to literature - but in my view, this would require a substantial effort in revising the paper.

I have included more specific comments below, but they should not be taken as an extensive list. My main recommendation to the authors would be to focus on making it much clearer to the reader what their key findings are and how they arrive at their interpretations.

Reply: We revised the manuscript as suggested, mainly by 1) reorganizing the results and discussion. 2) clarifying numerical quantities in detail, such as correlations, averages, and anomalies. 3) checking and revising any ambiguous or imprecise descriptions in the manuscript and reorganized sentences with unclear meaning. In

summary, we improved the text and reorganized the results and discussion so that the main findings and interpretations are clear.

**MAJOR COMMENTS**

There are many statements through the manuscript of 2020 being a year of very high sea ice exports through the gateways. The clearest example is perhaps L443, which states that area flux through the gateways was "extremely large compared to the 1988-2020 climatology". I find that this is an inaccurate description of the data as described. E.g. in figure 3: 2020 looks like a year of fairly high exports, but does not seem to particularly stand out; for example, area exports were higher in all three gates preceding year of 2019. Elsewhere, it is stated that the JFM 2020 area flux through the FS gateway was 1.2 times the 1988-2020 mean (L219) - this does not strike me as extreme given the large interannual variability (and there is a corresponding ~20% *negative* SIAF anomaly through the FJL-NZ gate in 2020). Similarly for "extremely low" chlorophyll concentrations in April (L20) - is this really substantiated by the analysis? I think it is important to be quite careful with language here; strong statements need to be supported by a clear justification based on underlying data. (Note: I do not think this season being less "extreme" means that it is not worth studying! For example, an observation that an extreme AO year only had a moderate impact on sea ice outflow would in my view be a valuable contribution.).

Reply: We checked the statement in the entire manuscript and revised it to "relatively large". In the section on SIAF anomalies, we reorganized such similar phrasing to ensure the accuracy of the data description (Line 550 and 566).

For the description of "extremely low" chlorophyll concentration, we gave a time series of spatially mean chlorophyll concentration over the Barents and Greenland Seas for April 2005–2020 (not shown in the revised version). In the southern Greenland Sea, spatially mean chlorophyll concentration is the lowest over the period 2010–2020. In the revised manuscript, discussion on chlorophyll concentration has been removed as suggested.

[Figure]

Figure 1. Mean chlorophyll concentration over the Barents and Greenland Seas in April for 2005–2020. The red dashed line indicates the chlorophyll concentration over the southern Greenland Sea in April 2020.

The Results and Discussion sections in my view need reorganisation. There are many instances where some fairly strong claims which are not based on the present analysis are included in Results. Conversely, much of the Discussion section consists of the presentation of whole new sets of results and analysis that have not previously appeared in the manuscript. I strongly suggest reorganising the manuscript such that the Results section is reserved for the presentation of the outcome of the analysis (including that of SST and chlorophyll), and the Discussion section for the authors' interpretation of the data/comparison with literature/qualified speculation etc. Given the complex topic and the many different datasets involved, the Discussion section of this paper is a great opportunity to carefully guide the reader through each key argument with reference to results from the different analyses.

An example of discussion in the Results section is found in ~L288 - L293. Here, there are some fairly broad statements about stratification which are not actually substantiated in the data. Similarly with e.g., L233-L235 / L239-241 / L299-L301 etc. These are important points but they require explanation and are not in my view obvious from the data alone.

An example of results in the Discussion section is the correlation analysis in 4.1. This reads very much like new results to me. Same with the SST/chlorophyll analysis that follows.

Reply: We reorganized the results and discussion sections. The main changes are:

1) We moved the correlation analysis between SIAF and AO/CAI to section 3.2 and revised the section title to "Anomalies in Arctic sea ice outflow and its link to atmospheric circulation patterns". (Line 286-296)

2) We moved the SST correlation analysis to section 3.4, because this section focuses on the anomalies in the BGS. We revised the title to "Anomalies in sea ice and sea surface temperature in the Barents and Greenland Seas". (Line 380-396)

3) Speculations in the results section, such as the connection between Arctic SIA and SIA of BGS as speculated by L233-L235, was removed as we calculated the correlation between the two and found it to be insignificant. We removed L239-241/L299-L301 to avoid duplication of descriptive results. For L288-L293, we removed the statement of oceanic stratification due to the absence of a strong explanation.

After reorganization, section 4.1 covers the impact of extreme atmospheric circulation patterns on dynamics and thermodynamic processes of sea ice before reached the Fram Strait; section 4.2 discusses factors other than Arctic sea ice inflow that affect sea ice anomalies in the BGS, such as air temperature, atmospheric surface heat fluxes, and NAO; section 4.3 discusses the sea ice variability in the BGS in years with high/low Arctic sea ice outflow, and whether the summer AO has an impact on the sea ice condition in the BGS.

If I have understood correctly, the atmospheric/sea ice area/chlorophyll analysis was done using averages across the area defined on L82 and shown as black polygons in Figure 1 BGS (if this is *not* the case, I suggest making it clearer which regions are used). The GS and BS boxes cover a vast area spanning quite different climatic environments and quite different ecosystems. I would guess that about half of this area is more or less never ice-covered in the present epoch; if chlorophyll in the southeastern part of the domain is influenced by sea ice inflows it may be in a rather indirect way. Likewise, if Figure 7 shows surface heat budget for the entire BGS area, I am not sure how meaningful they are. The authors need to justify this choice of a study area and discuss the implications of an analysis spanning widely different domains (or, if I have misunderstood, I would suggest that they clarify their methods..).

Reply: Yes, we used the averages within the black polygons in Figure 1 for these analyses. In the revised manuscript, we changed the east and west boundaries of the study region to the shoreline (Figure 1, Line 104), and recalculated the corresponding results. In conjunction with the later suggestion, we removed the discussion of chlorophyll. For the surface heat budget, we recalculated the surface heat flux anomalies and their changes in SIT, not over the entire BGS, but in the climatological ice-covered BGS. (Figure 8, Line 443-452 and 457-459)

I would recommend going through the manuscript in general for clarity. In quite a few instances, the meaning of a sentence is ambiguous or, I suspect, not in line with the intended meaning. I've included some examples below under "technical/language", but

they do not constitute a complete list.

Reply: We revisited the manuscript to avoid ambiguities and incorrect meanings.

I found the discussion of ecological conditions/phytoplankton to be severely lacking. First, it looks like the discussion is based on satellite chl-a concentrations from areas not directly influenced by sea ice (spring blooms in southern BS, eastern/southern GS). It was unclear to me whether the authors believe that conditions were favourable (L398) or unfavourable (L19) to biological production. Repeated statements about extremely low chlorophyll-a in April 2020 seem to refer to sea ice-covered areas where there are in fact no satellite measurements (Fig 9a). The statement that "phytoplankton seeding" and "residue of marine nutrients" were responsible for high primary production (L399-L400) is neither explained nor substantiated at all. I would suggest removing the chlorophyll discussion from the manuscript unless it is completely overhauled.

Reply: In the original manuscript, we quantified the relationship between chlorophyll and sea ice area over the BGS using satellite remote sensing products. Due to the limitations of the data, we attempted but could only do some simple analyses that could not substantiate and explain further mechanisms. As suggested, we removed the discussion of chlorophyll from the revised manuscript.

I would say something along the same lines about the section on surface heat fluxes (L304 - L323). First, there is the issue of (apparently) using integrated fluxes over areas including huge ice-free areas of the north Atlantic to assess ice melting in the small northern/westernmost portions the domain (I would expect there to be large heat losses from the ocean to the atmosphere in the southern BS and over the West Spristbergen Current in winter, for example - how do they affect this estimate?). Second, the values that are given for estimated ice thickness change spans a huge range (1 to 41 cm). Lastly, I think it would be quite helpful to plot the actual delta h alongside the actual fluxes to clarify what we are actually looking at.

Reply: 1) We recalculated the atmospheric surface heat flux anomaly over the climatological ice-covered BGS. (Figure 8, Line 443-452)

2) We then recalculated the estimated reduced ice thickness (Line 457-459) and superimposed it on the Figure 8. (Figure 8, Line 469)

**MINOR COMMENTS**

It might be nice to show time series of AO/CAI for context since these are pretty central to the paper. It's not strictly necessary, but I would suggest adding this as a small figure, at least in the supplementary.

Reply: We added the AO/CAI time series to the appendix. (Figure A1, Line 797)

It is stated in the abstract (L14) and elsewhere that "the variability of.. total SIAF was dominated by changes in ice motion speed", listing a high and significant correlation. In the text I could not find any details about how you actually calculated this, and given that this is a key point in the abstract I think the authors need to elaborate on actually how this quantity is computed, e.g.:

Is this a correlation between monthly values of a) SIAF across all three gates and b) mean speed across all three gates? Over what time period?

How are no-ice instances counted (for SIM and SIAF)? As zero, or are they not included?

Reply: We added the corresponding description in the methods section.

1) Detailed description of the calculation: "To quantify the relative contributions of changes in SIM and SIC to the variability of SIAF on a seasonal scale, we also calculated the correlation between the sum of the monthly SIAF and the mean SIM speeds/SIC through the three passageways for winter (JFM) and spring (AMJ) in 1988–2020." (Line 194-197)

2) Description of the no-ice instances: "Note that there is no SIM vector when the SIC is below 15% (Tschudi et al., 2019). In this case, the SIAF is ignored." (Line 175-176)

Along similar lines, it should be clearly stated whether "sea ice thickness anomalies" include "thickness" from ice-free periods. (Fine either way, but "thinner ice" and "less frequently ice-covered waters" are different things, for example).

Reply: In the calculation, SIT anomalies mean that SIT is thinner or thicker than the 2011–2020 average. We added sentences to clarify sea ice thickness anomalies. (Line 369, 371)

I think it should be discussed whether comparing trajectories from one single day (e.g. 31 May 2020) with a climatological mean vector field (e.g. May 1988-2020 mean drift) may (or may not) influence the analysis. Is a trajectory along an average field different from a time average of several trajectories? Does the long-time average introduce a low speed/short trajectory bias? (I don't really know - but I think it warrants at least a brief comment in the paper given that this is an important point in your study).

Reply: This may be a misunderstanding due to the lack of clarity in the labeling of Figure 4. Figures 4a-c all refer to the backward trajectory from a specific date in 2020 back to January 1. The 1988–2020 mean trajectories in Figures 4d-f do not refer to the temporal averaging of multiple trajectories, but rather the average SIM vector field in 1988–2020. We revised the label, for example, from "April 2020" to "April 30, 2020". (Figure 4, Line 345)

SIAF is basically the integrated product of SIM and SIC, so if it were not SIM that controlled SIAF variability it would presumably be SIC (or am I misunderstanding?).

You should therefore probably include the corresponding correlation between SIAF and SICs somewhere in the paragraph at L236 in support of your statement that SIM controls SIAF.

Reply: We calculated the correlation between SIAF and SIC in winter and spring, and added a description in section 3.2. (Line 261-262)

L45: I would revise the statement that sea ice outflow "contributes" to deep water formation in the north Atlantic. Please clarify what is meant here (presumably that high sea ice export -> high fw input -> increased stratification -> *inhibited* dw formation - or am I missing something?). I also can't see that Lemke et al. 2000 is a great reference here - they show a large variability in sea ice export but don't actually look at DW formation as far as I can tell?

Reply: We revised the sentence to "…significantly affects deep water formation in the north of the Atlantic Ocean.". (Line 50-52) We cited the following literatures that support the notion that freshwater from sea ice outflow can regulate the rate of deep water formation in the North Atlantic Ocean.

Dickson, R.R., Meincke, J., Malmberg, S.A.andLee, A.J.: The "Great Salinity Anomaly" in the Northern North-Atlantic 1968-1982, Prog. Oceanogr., 20(2): 103–151, https://doi.org/10.1016/0079-6611(88)90049-3, 1988.

Rahmstorf, S., Box, J.E., Feulner, G., Mann, M.E., Robinson, A., Rutherford, S.and Schaffernicht, E.J.: Exceptional twentieth-century slowdown in Atlantic Ocean overturning circulation, Nat. Clim. Change., 5(5): 475-480, https://doi.org/10.1038/NCLIMATE2554, 2015.

Figure 1 is excellent. The authors could consider adding mean SIC or similar (not necessary but might give some additional context to readers unfamiliar with the region).

Reply: We added the average SIC for January-March 2020 as a background. (Figure 1, Line 104)

The switch from CS2SMOS to PIOMAS seems to warrant a (quick) comparison of the two in the overlapping period. Should add at least a sentence about this somewhere around L123.

Reply: We compared the two SIT products during the overlapping period and added an explanation in the data section. (Line 141-149)

Which temperature does the oiSST give for fully ice-covered waters? What about for partial ice concentrations? Should be included in the presentation of this product somewhere after L124, and may be relevant for the interpretation of the SST data.

Reply: For fully ice-covered waters, the proxy SST is replaced by the freezing points

of seawater, which is defined using the climatological sea surface salinity (Banzon et al., 2020). We added a detailed description in the data section. (Line 155-158)

Banzon, V., Smith., T. M., Steele, M., Huang, B., and Zhang, H.-M.: Improved estimation of proxy sea surface temperature in the Arctic, J. Atmos. Oceanic Technol., 37, 341–349, https://doi.org/10.1175/JTECH-D-19-0177.1, 2020.

L159: I believe that there is no SIM vector for SIC<15% in this product? I don't believe that is a significant problem for your methods, but it is probably worth a mention here.
Reply: We added a mention in the methods section. (Line 175-176)

L161: Was one trajectory computed from each grid point on each gateway?
Reply: In fact, we divide a total of 400 points from the location of the three passageways and then calculate the backward trajectory from these points. The number of points varies for each gate, from 200 points (with a distance of ~ 448 km) for the FS, 100 points (with a distance of ~ 284 km) for S-FJL and 100 points (with a distance of ~ 326 km) for FJL-NZ.

L169: This seems to imply forward propagation, I would suggest reordering the equations to show backward propagation if that is what you do. Also, if delta t is negative as suggested in the preceding paragraph, I think this equation is in fact incorrect (wrong sign of second term)?
Reply: We revised the equation to show backward propagation, and the revised equation has a delta t of one day. (Line 208-209, 211)

L223. If you attribute the 20% positive anomaly at FS to the AO anomaly, it seems strange to say that 20% negative anomaly at FJL-NZ means that flow across this gate was "not sensitive". Could it not be a similar-magnitude response of opposite sign? Please elaborate.
L226: See above. This seems like a huge negative anomaly, and the idea that positive=response and negative=insensitivity needs an explanation at least. I also have a hard time seeing the 85% negative anomaly in Figure 3c - is this because the absolute values are small?
Reply: The responses is as follows:
    1) As mentioned in the result section, positive AO anomaly promotes sea ice transport to the BGS, so we assume that a response to positive AO anomaly implies an increased SIAF, which would be large than the 1988-2020 climatology. In contrast, the FJL-NZ passageway showed a large negative SIAF anomaly. We removed the description of response being sensitive or insensitive and revised the statement. (Line

271-273, and 278)

2) L226 refers to the fact that the spring cumulative SIAF through the FJL-NZ passageway is only 14.1% of the 1988-2020 climatology, while Fig. 3c is the difference between the monthly SIAF and the 1988-2020 climatology. So it is difficult to see the 85% negative anomaly there.

L230: The phrasing here is a bit ambiguous - should make it clear what exactly these percentages are. (I assume that they are the fraction of SIAF through all three gates that went through FS - but it can read as the percentage of the *anomaly*). Same for L13.
Reply: We revised the phrasing for clarity. (Line 17-18, 288-289)

L233: In my view, this is a strong statement that needs substantiation. What is your evidence that the low Arctic SIA was a result of increased outflow? Probably also a better fit in Discussion.
Reply: We calculated the relationship between Arctic SIA and SIAF and found that there is no significant correlation between them. So we removed the corresponding part.

L237: Please explain exactly what this correlation is.
Reply: We added a detailed explanation of this SIAF and SIM correlation calculations. (Line 194-197)

L251: " the anomalies of sea ice volume outflow.." This is a bit confusing. You don't have these numbers, right - is it that you would *expect* the volume export anomaly to be more pronounced than the area export anomaly? Rephrase for clarity.
Reply: We rewrote this sentence to "Although the source region of the ice was generally dominated by relatively thick ice, the SIAF anomalies in winter–spring 2020 was expected more obvious than the sea ice volume outflow anomalies as the negative trend of SIT in the Fram Strait (Sumata et al., 2023) is generally larger than that of SIC (Zamani et al., 2019)." But we finally removed this sentence considering the low relevance to the content of the manuscript.

Sumata, H., de Steur, L., Divine, D.V., Granskog, M. A., Gerland, S.: Regime shift in Arctic Ocean sea ice thickness, Nature, 615, 443–449, https://doi.org/10.1038/s41586-022-05686-x, 2023.

Zamani, B., Krumpen, T., Smedsrud, L.H., and Gerdes R.: Fram Strait sea ice export affected by thinning: comparing high-resolution simulations and observations, Clim. Dyn., 53, 3257–3270, https://doi.org/10.1007/s00382-019-04699-z, 2019.

L267-L268: This seems true for the FS, but not for the two other gates?

Reply: This sentence is rewritten to state that it only applies to the FS. (Line 339-340)

Figure 4: I suggest using much thinner lines to show that these are trajectories and not a continuous scalar field - I think this figure is nice, but it took me quite a while to figure out what was going on.. Given that trajectories don't run past January, I don't think the colour by date actually adds that much, so I wouldn't be worried if thinner lines show the colours less well. Should also change the labels in abc from e.g. "April 2020" to "30 April 2020" if you indeed only show back-trajectories from one single day in these. (You should also explain why you chose to compare one day per month with monthly climatological means).

Reply: 1) We tried to use a thinner line to show the trajectories, but we found that the display is still dense, probably because we used 400 points located at three gates as endpoints. 2) The trajectories in Figure 4 all refer to the backward trajectory from a particular day. Therefore, we changed the labels of Figure 4 to specific dates. (Figure 4, Line 345)

L298: It's not clear to me from Figure 6a that there was a positive SIT anomaly in the BS - please explain.

Reply: We revised this sentence to show that both positive and negative SIT anomalies in the Barents Sea were small. (Line 370)

L301: Similarly, it does *not* look to me like an overall positive anomaly in the GS in Figure 6d. And it seems strange not to address the east-west pattern in the GS here. And, as mentioned elsewhere, I think you also need to say something about whether the positive anomalies are a result of anomalously high sea ice *extent* or anomalously thick ice since this could impact your argument about increased import of thicker ice from the central Arctic (probably depends on how you compute these means).

Reply: We recalculated the SIT anomalies in the BGS and added a discussion about the east-west pattern of SIT anomalies (Figure 6, Line 376-379) and explained the meaning of the SIT anomalies. (Line 369, 371)

L337-8: Is it surprising that you did *not* find a significant correlation between SIAF and AO in months other than February? Seems to warrant at least a mention.

Reply: We revised this sentence to "During January–June, there is a significant positive correlation between SIAF and the AO identified in February, but not in other month" (Line 291-292)

L340: What about R?

Reply: We added the specific value of $R$ to this sentence. (Line 294)

L345-351. I found it difficult to see this in Figure A1ab, please explain how you arrive at this conclusion. To me, it looks like there is little difference except perhaps that the drift is stronger in the negative AO phase (A1b). It also looks like these are the trajectories of ice arriving at the gate in mid-summer - so is it actually representative of AO anomalies in winter? Please clarify.

Reply: We revised this paragraph to elaborate. The specific response is as follows:

1) This can be seen at the end of the backward trajectory in Fig. A2 ab (blue trajectory). The end of the blue trajectory in Fig. A2a extends westwards, while that in Fig. A2b moves closer to the prime meridian. This suggests that winter AO+ is more conducive to sea ice outflow from the central Arctic Ocean to the BGS (Rigor et al., 2002). (Line 409-410)

2) These represent the backward trajectory of ice arriving in the Fram Strait on 30 June back to 1 January, with emphasis on the location of the winter (JFM) trajectory. We chose years with winter AO anomalies to check whether the January–June backward trajectory of sea ice arriving in Fram Strait in June is affected by winter AO anomalies.

Rigor, I.G., Wallace, J.M., and Colony, R.L.: Response of sea ice to the Arctic Oscillation, J. Clim., 15(18), 2648–2663, https://doi.org/10.1029/1999gl002389, 2002.

L365: Are these SST anomalies integrated over the whole area? Also, are the maps in Figure 8 really anomalies from the climatological mean for the month? To me they look like anomalies from the SST average across the entire season or something (+4C anomalies in the southern BS in July and -3C anomalies in the northernmost range in April both strike me as weird). I could certainly be wrong - but it might be a good idea to double check.

Reply: We checked the calculation and found that the previous result was wrong. Therefore, we redrew original Figure 8 (changed to Figure 7 in the revised version). Figure 7 does not show the +4C anomalies in the southern Barents Sea in July. (Figure 7, Line 400)

L389 onward: Again, it seems strange to me to do this sort of analysis across this very large area - you at least need to discuss the implications of 1) including large perennially ice-free areas, and 2) there being no satellite measurements of chlorophyll in ice.

Reply: Considering the major comments above, we acknowledge that the discussion of chlorophyll is inadequate. And we removed the chlorophyll discussion from the manuscript.

I found the Conclusions section to be quite clarifying and well-written. I would

encourage the authors to use a similar style in an updated Discussion section.

Reply: In the revised discussion, we used a similar style for clarity.

**TECHNICAL/LANGUAGE**

L13: "77.6%": I gather from the conclusion that this is the fraction of the total flux through the three gateways that goes through the FS gate. That was not clear to me when reading the abstract alone.

Reply: We revised the sentence to "…The relatively large total SIAF, which was dominated by that through the Fram Strait (77.6%) …". (Line 17-18)

L51: "More primary productivities" - please use more precise language.

L55: I would suggest reordering this sentence for clarity.

L63: "Relatively extreme" - please use more precise language.

Reply: We revised these sentences for clarity. (Line 57, 63-65, and 71)

L69: "Thereby.. ..the BGS." - Meaning of this sentence is unclear.

Reply: This sentence is rewritten for clarity. (Line 77-79)

L82: Studies -> Study?

L107: "abnormal" - "anomaly"?

L124: "Could be used as the best proxies for.." - please rephrase. "We use SST and chl as proxies for.."?

L137: Suggest replacing "a significantly more advanced" with "an advanced" or similar ("more" relative to what?).

L143: "According to" - "In accordance with" or similar?

L146: These directions aren't really S/N? Maybe replace "southward" with "toward the BGS" or something.

Reply: We corrected these grammatical mistakes and inappropriate expressions as suggestions. (Line 97, 123, 150-151, 165, 174, and 176-177)

L185: "with the values maintaining the top three.." - This phrasing is a bit unclear (here and elsewhere in the manuscript).

Reply: We checked the entire manuscript and changed all similar phrasing to "ranged XX" or "rank XX" for clarity. (Line 225 and 514)

L215-L217: Suggest moving this sentence to data/methods (and adding e.g. "relative to the 1988-2020 climatology" after "June 2020" in L218).

Reply: This sentence is moved to the methods section (Line 191-194). We rewrote most

of section 3.2, and added "relative to the 1988-2020 climatology" where appropriate. (Line 264)

L238: Maybe "monthly SIM speed" to avoid confusion.
L256: "very analogous to" - "similar to"?
Reply: Revised phrasing to avoid confusion. (Line 258 and 306)

L261: "confluence of the Kara Sea and the CAO" - please explain.
Reply: The labeling of the Kara Sea is added to Figure 4 in order to indicate the meaning of this sentence. (Figure 4, Line 345)

L262: "exhibited a relatively high tortuous feature" - unclear-please explain.
Reply: We revised this sentence to "…were curved than that from the Fram Strait." (Line 332-333)

L280: Sentence needs fixing; meaning of "since then" is unclear.
Reply: We rewrote this sentence to "SIA in the BGS generally reaches its annual maximum in April each year, and then begins to decline as the air and ocean temperature rises". (Line 351-352)

L280-L288: I strongly suggest revisiting this entire section for clarity; I found it quite difficult to follow most of it.
Reply: We reorganized and rewrote this section for clarity. (Line 357-361)

L287: What is this correlation? Is it between the sum of the SIAF through all three passageways and SIA in the BGS? Also: "has been identified" should be rephrased for clarity (e.g. "we found a correlation.." if that is the meaning).
Reply: We rewrote the sentence to clarify that it refers to the correlation between the sum of SIAF through the three passageways and the SIA in the BGS. (Line 362-364)

L304: "dominated" - "governed"?
L337: "most active" - rephrase for clarity.
L347: "spatial scope" - "spatial extent"?
L376: typo (southern/northern).
Reply: We corrected misspellings and inappropriate phrases, and rephrased inappropriate expressions (Line 391). For Line 304, 337 and 347, we deleted these expressions.

L396: "a single SST" - rephrase for clarity.

Reply: In the revised version, we removed the chlorophyll section, so this sentence was also removed.

I suggest looking over sentences starting with "thereby", "furthermore" etc. and checking that these preserve the intended meaning.
Reply: We checked the entire manuscript and revised these sentences to ensure that the intended meaning was retained. (Line 568 and 573)

Figure A1: Please label clearly which plots correspond to positive/negative phases, or at least explain/refer to labels in the caption ("AO+sd" is not very intuitive on its own). I would also suggest adding the number of years that go into the mean (e.g. "n=3") somewhere on each subpanel (not necessary but would aid interpretation).
Reply: We added a description in the caption and the number of years in the subpanel. (Figure A2, Line 799)

**Response to RC4**

We express our sincere appreciation for the time and valuable feedback provided by the reviewer on the manuscript titled "Impacts of anomalies in Arctic sea ice outflow on sea ice in the Barents and Greenland Seas during the winter-to-summer seasons of 2020". We carefully reviewed all comments and revised them to ensure that our manuscript was significantly enhanced.

**General comments:**

1. What is the reason to choose these two atmospheric circulation patterns for assessing the effects of large-scale atmospheric circulation on the changes in Arctic sea ice outflow? I would suggest that the authors also look at whether there is an abnormal North Atlantic Oscillation in 2020, and if so, the influence of the North Atlantic Oscillation on Arctic sea ice outflow should be discussed.

Reply: We chose AO and CAI mainly because they showed obvious positive anomalies in winter-spring 2020, ranking high in 1979–2020. While NAO did not show such strong anomalies in 2020. In the revised manuscript, we added a discussion of the effects of NAO on the BGS (Line 460-468), quantifying the relationship between NAO and sea ice area.

2. Section 3.3 focus on the comparison of the reconstructed sea ice backward trajectories in 2020 with the 1988-2020 climatology. The comparison assumes that the reconstructed sea ice backward trajectories are convincing. However the validation of the reconstructed backward trajectory method is not sufficient. I would suggest that the authors provide more assessments on the validity of the reconstructed trajectories using buoy observations.

Reply: We added a discussion on verifying the reliability of reconstructed ice backward trajectories using buoys obtained from MOSAiC and International Arctic Buoy Program. In the AO+ and CAI+ cases, the Euclidean distances between the reconstructed trajectories and buoys trajectories were smaller than in the AO– and CAI– cases. But the cosine similarities were all above 0.9 in the AO and CAI cases. This indicates that the ice drift orientations obtained from the reconstructed trajectories regardless of the phases of AO and CAI were similar to the buoys and were reliable. (Line 309-322)

3. Section 3.4 discussed the anomalies of the sea ice area and thickness in the Barents and Greenland Seas. The data analysis on sea ice area is relatively adequate. However the analysis of sea ice thickness anomalies is mostly qualitative. I would suggest that

the authors provide more quantitative results on sea ice thickness anomalies and discuss them in details.

Reply: We calculated quantitative results for the SIT anomalies. We found that the SIT anomalies in the Greenland Sea in March–April exhibited an east-west pattern. (Line 376-379)

4. If the abnormal Arctic Oscillation and Arctic sea ice outflow do not occur in winter, but in other seasons, will the effect be different for the ice and marine environment conditions in the Barents and Greenland Seas? It would be better to add more discussion on this.

Reply: The discussion of the impact of positive summer AO (JAS) on the BGS is added in section 4.3. In 2016, we found that AO+ also contributed to the summer Arctic sea ice outflow to some extent. However, the SIA of the BGS was even smaller than that estimated form the linear regression of 1979–2020. (Line 528-532)

5. It is a last resort to use different sea ice thickness products (radar altimeter and PIOMAS model-based data) in different seasons. My concern is whether using different data produces inconsistent results. For example, during the freeze-up period, whether there is deviation or even contradiction between the qualitative conclusion and quantitative results using PIOMAS model-based data and radar altimeter?

Reply: We calculated the SIT anomalies for these two SIT products from December to April. The results showed that the spatially averaged differences in SIT anomalies between them are about 6.0%-13.3% of the monthly magnitude. The correlation between the SIT anomalies calculated using the two dataset is 0.95 in 2011-2020 ($P<0.05$). We believed thus that the difference between these two datasets is acceptable. (Line 141-149)

**Specially comments:**

1. Line 72, change "during winter-spring 2020" to specific month, since you do not define the range of winter and spring months before that.

Reply: We changed the sentence to "winter (JFM)–spring (AMJ) 2020". (Line 83)

2. Line120-122, "We regridded the monthly SIT data on the 25-km EASE-Grid to maintain consistency with the CryoSat-2/SMOS SIT data." These two datasets also have different temporal resolutions. How was this difference addressed in your study?

Reply: We added the description and revised this sentence to "We regridded the monthly SIT data on the 25-km EASE-Grid and calculated the monthly average CryoSat-2/SMOS SIT data to maintain the spatial and temporal consistency of the two SIT dataset.". (Line 139-141)

3. Line 161, change "restructured" to "reconstructed" to unify the expression and apply to the entire manuscript.

Reply: We changed "restructured" to "reconstructed" throughout the manuscript. (Line 199, 203, 206, 306, and 422)

4. Line 271, "enhanced sea ice meridional motion", remove the "meridional", since you do not directly calculate the meridional sea ice motion speed.

Reply: We removed the "meridional". (Line 343)

5. Line 294, The text on the right side of Figure 5 is too busy and not intuitive enough. It is preferable to express the trend of sea ice area graphically.

Reply: We used a bar chart to depict the trends of the sea ice area. (Figure 5, Line 365)

6. Line 319, "The anomalies in cumulative surface heat fluxes from January to June 2020 can be related to a reduced decrease of 0.01–0.41 m in SIT, estimated using the Eq. 4" This is ambiguous. Does cumulative mean cumulative over time or across net surface heat fluxes?

Reply: We recalculated the changes in SIT using the atmospheric surface heat flux anomalies over the climatological ice-covered regions and rewrote the sentence. (Line 457-459)

7. Line 327, change the y-axis title in Figure 7 to "Surface heat fluxes anomaly", because it is surface heat fluxes anomaly instead of surface heat fluxes in the figure caption.

Reply: We changed the y-axis title to "Atmospheric surface heat fluxes anomaly". (Figure 8, Line 469)

8. Line 377-378, "In addition, we examined the statistical relationship between the April SIA and the monthly SST with a lag of 1–3 months in the BGS (Table A4)." Is this correlation for the detrended SIA and SST? This comment applies to the entire manuscript.

Reply: We added notes of detrended to the entire manuscript. (Line 393)

9. Line 388, "The Chl-a over the southern Greenland Sea in April 2020 was smaller compared to the previous 5 years." Give the latitudinal range of the southern Greenland Sea.

Reply: We revised the sentence to "The *Chl-a* over the southern Greenland Sea (72°– 76°N) in April 2020 was smaller compared to the previous 10 years.", and the following figure supports this conclusion. We finally accepted RC3's suggestion to remove the

part of chlorophyll concentration.

[Figure]

Figure 1. Mean *Chl-a* concentration over the Barents and Greenland Seas in April for 2005–2020. The red dashed line indicates the *Chl-a* concentration over the southern Greenland Sea in April 2020.

10. Line 389, "A significant negative correlation between Chl-a and SIA in April over 1998–2020 was identified". The geographical scope of this sentence is unclear. Is there a negative correlation between Chl-a and SIA in the BGS or only in the Greenland/Barents Sea?

Reply: We revised the sentence to "In the southern Greenland Sea, a significant negative correlation ($R = –0.44$, $P < 0.05$) was found between *Chl-a* and SIA in April over 1998–2020". The section on chlorophyll concentration was not included in the revised manuscript because we removed this section.

11. Some illustrations need to be further revised, for example, the definition of the study area in Figure 1 is not so normative.

Reply: In order to rationalize Figure 1, we redrew it so that the northern boundary of the study region could be consistent with the defined passageways and the bordering areas of Greenland and other islands could be aligned with the shoreline. Based on the revised study region, we updated the results. (Figure 1, Line 104)